# DIFFERENTIALLY PRIVATE RANGE SUBGRAPH COUNTING

## ABSTRACT

Subgraph counting is a fundamental problem in graph analysis. Motivated by the practical need to perform graph analytics on subgraphs defined by selected vertices (or edges) rather than the entire graph, as well as privacy concerns, we initiate the study of private range subgraph counting. Given an $n$-vertex graph $G$, where each vertex (or edge) has a $d$-dimensional attribute vector, a pattern graph $H$, and a set $Q$ of range queries $q$, our goal is to count the occurrences of $H$ in the subgraph of $G$ induced by vertices (or edges) whose attributes fall within $q$, all while preserving privacy. We give the first $\varepsilon$-differentially private algorithm for range subgraph counting, achieving near-optimal accuracy (up to a polylog-arithmic factor of $n$) for constant privacy parameter $\varepsilon$ and dimension $d$, with no additional computational overhead compared to non-private algorithms. We also demonstrate that by relaxing to $(\varepsilon, \delta)$-DP, we can achieve smaller additive errors. Furthermore, our results generalize the subgraph counting results of the partially dynamic model in (Fichtenberger et al., 2021). Empirical evaluations demonstrate that our algorithm significantly outperforms baseline methods in accuracy while ensuring strong privacy guarantees.

## 1 INTRODUCTION

*Subgraph counting* is essential for understanding the properties of a data graph and has been extensively studied (Alon et al., 1995; Bera et al., 2021; Björklund et al., 2014; Chiba & Nishizeki, 1985; Curticapean et al., 2017; Assadi et al., 2019; Fichtenberger et al., 2020). Given a *host* graph $G = (V, E)$ and a *pattern* graph $H$, a subgraph of $G$ that is isomorphic to $H$ is called an occurrence of $H$. The goal of subgraph (or pattern) counting is to determine the number of occurrences of $H$ in $G$. Subgraph counting is a key graph statistic; for instance, counting triangles and $k$-stars is crucial for computing the clustering coefficient, which is valuable for evaluating the effectiveness of friend recommendation systems. Counting 4-cycles, closed loops of four nodes, is particularly useful for measuring clustering tendencies in bipartite graphs, such as those found in online dating platforms or mentor-student networks.

In many applications, beyond counting subgraphs in the entire graph, we are often interested in counting subgraphs within specific subgraphs. This is driven by practical demands for performing graph analytics on subgraphs relevant to selected vertices (or edges) rather than the entire graph. For instance, in patient networks, we may be interested in counting patterns within the subgraph induced by patients of similar age or geographic location. These subgraphs can be defined based on specific age ranges, geographic areas, or other relevant attributes. In financial networks, counting transaction patterns among entities with similar risk profiles or locations can help identify fraudulent activities or assess systemic risks within the financial system. Another example involves relational event graphs (Bannister et al., 2013). In this context, we are given a graph $G = (V, E)$, where each edge $e \in E$ is associated with a real-valued timestamp. We may wish to count the occurrences of certain patterns within a specific time range, which corresponds to the subgraph induced by all edges that fall within that time frame.

Now we formally introduce the *Range Subgraph Counting* problem that addresses the pattern counting scenarios discussed above.

**Definition 1.1** ((Vertex-attributed) range subgraph counting problem)**.** *Let $G = (V, E)$ be an undirected graph, where each vertex $v \in V$ has a real-valued attribute $\mathbf{a}(v) \in \mathbb{R}^d$. For a given interval $q = [\ell_1, r_1] \times \cdots \times [\ell_d, r_d]$, define $V_q = \{v \in V \mid \ell_i \leq \mathbf{a}_i(v) \leq r_i, i \in [d]\}$, and let $G_q$ denote the*

*subgraph of $G$ induced by $V_q$, i.e., $G_q = G[V_q]$. Let $Q = \{q = [\ell_1, r_1] \times \cdots \times [\ell_d, r_d] \mid \ell_i, r_i \in \mathbb{R}, \ell_i \leq r_i, i \in [d]\}$ be the query set.*

*Let $H$ be a fixed, connected pattern graph with $O(1)$ vertices. For each query defined by the interval $q$, the goal is to return the number of occurrences of $H$ in $G_q$. The pattern $H$ is fixed for all queries.*

Note that the attributes of the vertices may represent factors such as age or location, depending on the practical context. Additionally, an occurrence is only counted if all its vertices are contained within $V_q$; any occurrences involving vertices outside of $V_q$ are disregarded. We also study a variant of this problem in the setting where each edge $e$ has an associated real-valued attribute $\mathbf{a}(e)$, referred to as the edge-attributed range subgraph counting problem (see Appendix F). Furthermore, we note that our edge-attributed range counting strictly generalize the partially dynamic DP subgraph counting under continual observation as studied in (Fichtenberger et al., 2021). In the partially dynamic setting, the edge attribute is timestamp and only allow either insertions or deletions of edge. See Section 1.1 for more discussions. We remark that Deng et al. (2023b) studied the 1-dimensional (vertex-attributed) range subgraph counting and listing problems, focusing on optimizing the trade-off between space and query time.

While one could release the exact pattern counts in response to each query, it is important to recognize that the range subgraph counting algorithm lacks formal privacy guarantees, making it potentially "unsafe" from a privacy perspective.

In this work, we approach the range subgraph counting problem from the perspective of *differential privacy (DP)*. DP ensures that, even if there is a one-element difference in the database, the output of the algorithm remains statistically similar (see Definition 1.2). This means that DP algorithms allow for statistical analyses of sensitive individual data while guaranteeing that no specific individual's information is leaked (Dwork et al., 2006). When DP is applied to graphs, it can be divided into two types:edge-DP and node-DP. In the former, two adjacent graphs differ only by one edge, while in the latter, two adjacent graphs differ by one node and all the neighboring edges. In our work, we focus on edge-DP. Given two graphs $G, G'$ with the same set of nodes $V(G) = V$, we say $G, G'$ are *neighboring*, denoted by $G \sim G'$, if they differ in exactly one edge.

**Definition 1.2** (Edge DP (Dwork et al., 2006; Nissim et al., 2007))**.** *Let $\varepsilon > 0$ and $\delta \in [0, 1)$. A randomized algorithm $\mathcal{A}$ is $(\varepsilon, \delta)$-differentially private(DP) if for all events $S$ in the output space of $\mathcal{A}$ and all neighboring graph $G \sim G', \Pr[\mathcal{A}(G) \in S] \leq e^\varepsilon \Pr[\mathcal{A}(G') \in S] + \delta$. When $\delta = 0$, we say $\mathcal{A}$ preserves pure differential privacy (denoted by $\varepsilon$-DP). When $0 < \delta < 1$, we say $\mathcal{A}$ preserves approximate differential privacy.*

While DP has been extensively studied for subgraph counting in the entire host graph (see Section 1.1), private algorithms for range subgraph counting remain unexplored. The challenge with DP range subgraph counting arises not only from the high sensitivity, which is already present in standard DP subgraph counting, but also from the increased complexity of the queries. In range subgraph counting, each query is defined over a specific subgraph induced by a subset of vertices, making the problem more difficult as the algorithm need to handle multiple induced subgraphs efficiently while ensuring privacy.

Before presenting our main results, we outline a straightforward approach to achieve differential privacy (DP) in range subgraph counting: For each query $q \in Q$, compute the induced subgraph $G_q$, count the occurrences of the pattern graph $H$ (e.g. triangles), add Laplace noise to the counts, and return the noisy results. However, this approach has significant drawbacks. Specifically, it results in substantial additive error. The sensitivity of triangle counting in any specific graph $G_q$ is $\Theta(|V_q|)$, necessitating Laplace noise of $\Theta(|V_q|)$. According to the DP composition theorem (Dwork et al., 2006), this leads to a cumulative error of $O(|Q|n)$ when aiming for $\varepsilon$-DP (and $O(\sqrt{|Q|}n)$ for $(\varepsilon, \delta)$-DP). When $|Q| = \Omega(n^2)$, the resulting error becomes prohibitively large, rendering the results practically unusable. For example, in the case of triangles, where the total number of triangles in a graph is $O(n^3)$, the excessive error $O(n^3)$ for $\varepsilon$-DP becomes trivial. Furthermore, we note that range subgraph counting is a *nonlinear* problem, making it more challenging, and preventing the direct application of previous DP algorithms designed for linear queries. For instance, the sum of the number of triangles in two graphs is not equal to the number of triangles in their union.

**Our Contribution** We present the first efficient range subgraph counting algorithm that satisfies DP with nearly-optimal additive error, where an algorithm is said to be *efficient* if it runs in poly-

nomial time. We let $f_H(G)$ denote the number of occurrences of $H$ in $G$, and let $\mathrm{GS}_{f_H}$ denote the *global sensitivity* of subgraph counting of $H$ (see Definition 2.1).

**Theorem 1** (Pure DP (Vertex-Attributed) Range Subgraph Counting)**.** *For any $\varepsilon > 0$ , there exists an $\varepsilon$-differentially private efficient algorithm that, given a graph $G = (V, E, \mathbf{a})$, where the attribute of each* **vertex** *is a $d$-dimensional vector, pattern graph $H$, a query set $Q$, outputs all range subgraph counting queries which satisfy*

$$\max_{q \in Q} \left| f_H(G_q) - \widetilde{f}_H(G_q) \right| = O\left( \frac{\mathrm{GS}_{f_H} \cdot d \cdot \log^{3d+0.5} n}{\varepsilon} \right)$$

*with probability at least $1 - \frac{1}{n}$.*

If we relax the requirements to approximate DP, we can derive an algorithm with a smaller additive error, as stated in the following theorem.

**Theorem 2** (Approximate DP (Vertex-Attributed) Range Subgraph Counting)**.** *For any $\varepsilon > 0$ and $0 < \delta < 1$, there exists an $(\varepsilon, \delta)$-differentially private efficient algorithm that, given a graph $G = (V, E, \mathbf{a})$, where the attribute of each* **vertex** *is a $d$-dimensional vector, pattern graph $H$, a query set $Q$, outputs all range subgraph counting queries which satisfy*

$$\max_{q \in Q} \left| f_H(G_q) - \widetilde{f}_H(G_q) \right| = O\left( \frac{\widetilde{\mathrm{HS}}_{f_H}(G) \cdot d \cdot \log^{3d+0.5} n}{\varepsilon} \right)$$

*with probability at least $1 - \frac{1}{n}$, where $\widetilde{\mathrm{HS}}_{f_H}$ denotes the output in Algorithm 7.*

In the above, the quantity $\widetilde{\mathrm{HS}}_{f_H}$ can be viewed as an approximation of the higher-order local sensitivity (see (Nguyen et al., 2023)). The parameter $\delta$ is typically set to a value on the order of the reciprocal of a polynomial in the input size (e.g., $n^{-O(1)}$). It is implicitly incorporated within $\widetilde{\mathrm{HS}}_{f_H}(G)$, which exhibits a dependency on $\mathrm{poly}(\log(1/\delta))$. In real-world graphs, which are typically sparse, $\widetilde{\mathrm{HS}}_{f_H}(G)$ is often significantly smaller than $\mathrm{GS}_{f_H}$. For instance, when $H$ is a triangle, $\widetilde{\mathrm{HS}}_{f_H}(G) \approx d_{\max}(G) \ll \mathrm{GS}_{f_H} = n - 2$, where $d_{\max}(G)$ represents the maximum degree of graph $G$. The proof and detailed description of Theorem 2 can be found in Appendix D.1.

We also show that for the edge-attributed range subgraph counting problem, one can obtain an efficient pure DP (approximate DP) algorithm with the same additive error as the above. We present the formal statement Theorem 3 and give its proof in Appendix F.

We note that simply reporting the number $f_H(G)$ of subgraphs $H$ in the entire host graph while satisfying $\varepsilon$-DP incurs an additive error of at least $\Omega(\mathrm{GS}_{f_H})$. This is due to the fact that the additive error for the counting problem cannot be lower than the global sensitivity in the worst case (Dwork et al., 2006). Therefore, our upper bounds achieve nearly optimal additive error up to a factor of $\mathrm{poly} \log n$ for any constant $d$ and $\varepsilon$. Furthermore, note that our theorems still provide non-trivial bounds when $d$ is not necessarily constant but remains relatively small (e.g., $d = o(\sqrt{\log n})$). An interesting open question is how to obtain better bounds for higher dimensions $d$ (e.g. $d = \Omega(\log n)$).

Furthermore, we observe that the global sensitivity $\mathrm{GS}_{f_H}$ can be bounded to be $O(n^{2\rho(H)-2})$, where $\rho(H)$ is the *fractional edge cover number* of $H$ (Appendix A). Suppose $d, \varepsilon$ are constant. Then if $H$ is triangle, then $\rho(H) = 3/2$, which implies our DP algorithm for range triangle counting has error[1] $\tilde{O}(n)$; if $H$ is $k$-clique (i.e., a complete graph on $k$ vertices) or a $k$-cycle (i.e., a cycle with $k$ vertices), then $\rho(H) = \frac{k}{2}$, which implies an error $\tilde{O}(n^{k-2})$. The latter also implies for $k = 2$, i.e., $H$ being an edge, then the additive error is $\tilde{O}(1)$.

We experimentally test our DP algorithms for range subgraph counting on real network datasets in Section 4.

**Technical Overview** To design DP algorithms for the range subgraph counting problem with small additive error, we observe that many range queries overlap, making it unnecessary to add noise to each query separately. Our approach maps the graph's vertices to points in a $2d$-dimensional Euclidean space, based on vertex or edge attributes, translating the range subgraph counting problem into estimating the weighted sum of points within corresponding rectangles. Here, the weight of a

---

[1] $\tilde{O}(\cdot)$ hides polylogarithmic factors.

point reflects the number of occurrences involving the corresponding vertex pair. We employ a range tree data structure (Bentley & Saxe, 1978) to iteratively summarize these weighted sums within chosen ranges, adding Laplace noise to the weights of each node in the tree. To answer a range query, we traverse the tree to find the relevant nodes for the queried range. This approach effectively leverages query correlations, reducing the amount of noise required.

Our work shares similarities with the DP interval (and rectangle) query problem (see, e.g., (Dwork et al., 2015)), which focuses on reporting the number of points in a specified interval, often solved using a range tree. However, there are several key differences. First, we address edge-DP in graphs, whereas (Dwork et al., 2015) focuses on differential privacy in tabular data, where each row corresponds to an individual. Second, unlike point counting, our subgraph counting problem is nonlinear; specifically, the sum of occurrences of a pattern graph in two graphs is not necessarily equal to the number of occurrences in their union. Third, in our setting, a single edge change can affect many mapped points and significantly impact subgraph counts (e.g., one edge may participate in $\Theta(n)$ triangles). We address the latter two differences by employing a subgraph projection technique that uniquely maps each occurrence of a pattern graph $H$ to a distinct point in Euclidean space. This transformation allows us to appropriately apply the rectangle query algorithm to our problem.

## 1.1 RELATED WORK

**DP Subgraph Counting** The DP subgraph counting problem is a significant topic that has been extensively studied, primarily for the entire graph $G$. Nissim et al. (2007) improved the utility guarantees for triangle counting in differential privacy by incorporating instance-specific noise. Karwa et al. (2011) extended the smooth sensitivity approach to $k$-stars and proposed methods for computing local sensitivity to perform $k$-triangle counting. Kasiviswanathan et al. (2013) introduced a triangle counting algorithm under the node-DP framework. Zhang et al. (2015) developed ladder functions for various subgraph counting tasks. Nguyen et al. (2023) focused on optimizing run-time by calculating approximate smooth sensitivity for graphs with certain properties, achieving both privacy and utility while reducing time complexity. Additionally, several studies have examined subgraph counting under the local DP model, such as (Imola et al., 2021; 2022a;b; Eden et al., 2023). (Fichtenberger et al., 2021) studied DP subgraph counting in dynamic model, while our work explores subgraphs induced by vertices or edges whose attributes fall within specified ranges. For Vertex-attribute Range Subgraph Counting, the two problems are fundamentally different and incomparable. In the context of Edge-attribute Range Subgraph Counting, our work generalizes the partially dynamic problem in their work, where their problem becomes a special case of ours when treating edge timestamps as attributes. Instead of focusing on specific pattern graphs like triangles and $k$-stars, our approach generalizes to arbitrary constant-size pattern graphs.

**Differentially Private Range Queries** Muthukrishnan and Nikolov (Muthukrishnan & Nikolov, 2012) present algorithms for the half-space range counting problem under differential privacy, achieving good approximate accuracy in terms of average squared error. Deng et al. (Deng et al., 2023a) propose an algorithm for counting queries and bottleneck queries on shortest paths while ensuring differential privacy. A closer examination of their model reveals that they effectively address a range counting problem on a graph. A cut query on a graph is a specialized form of range counting, where the range space includes all possible cuts. The cut query problem is widely studied in the field of differential privacy, with significant research dedicated to it (Gupta et al., 2010; 2012; Dalirrooyfard et al., 2024; Blocki et al., 2012; Arora & Upadhyay, 2019; Eliáš et al., 2020).

## 2 PRELIMINARIES

Let $G = (V, E, \mathbf{a})$ be a weighted graph with node set $V$ of size $|V| = n$, edge set $E$ of size $|E| = m$ and vertex attribute vector $\mathbf{a} : V \to \mathbb{R}^d$. $H = (V_H, E_H)$ is a pattern graph such as $k$-star, triangle and so on. For simplicity, we let $V = [n] := \{1, 2, \ldots, n\}$. A subgraph of $G$ isomorphic to $H$ is called an *occurrence* of $H$. We use $f(\cdot)$ represents a function and use $f_H(G)$ to represent the number of occurrences of $H$ in $G$. For $\mathbf{x} \in \mathbb{R}^k$, we denote $\|\mathbf{x}\|_1 = \sum_{i \in [k]} |\mathbf{x}_i|$.

**Differential Privacy** The global sensitivity of a function is defined as follows.

**Definition 2.1** (Global Sensitivity (Dwork et al., 2006))**.** *For any function $f : \mathcal{X} \to \mathbb{R}^k$ defined over a domain space $\mathcal{X}$, the* global sensitivity *of the function $f$ is defined as* $\mathrm{GS}_f = \max_{G \sim G'} \|f(G) - f(G')\|_1$.

We will make use of the following post-processing theorem and basic composition theorem of differential privacy.

**Proposition 2.2** (Post-processing theorem (Dwork et al., 2006)). *Let $M: \mathbb{R}^{d_1} \to \mathbb{R}^{d_2}$ be an $(\varepsilon, \delta)$-differential private mechanism and let $h: \mathbb{R}^{d_2} \to \mathbb{R}^{d_3}$ be an arbitrary function. Then, the function $g \circ M: \mathbb{R}^{d_1} \to \mathbb{R}^{d_3}$ is also $(\varepsilon, \delta)$-differentially private.*

**Proposition 2.3** (Basic composition theorem (Dwork et al., 2006)). *For any $\varepsilon, \delta > 0$, the composition of $k$ $(\varepsilon, \delta)$-differentially private algorithms is $(k\varepsilon, k\delta)$-differentially private.*

**Laplace distribution and Laplace mechanism** We now introduce the definitions of Laplace distribution and Laplace mechanism.

**Definition 2.4** (Laplace distribution). *We say a zero-mean random variable $X$ follows the* Laplace distribution *with parameter $b$ if the probability density function of $X$ follows* $\mathrm{Lap}(b) = \frac{1}{2b} e^{-\frac{|x|}{b}}$.

**Fact 2.5.** *If $Y \sim \mathrm{Lap}(b)$, then $\Pr[|Y| > tb] \leq e^{-t}$.*

The sum of multiple variables that follow the Laplace distribution satisfies the following properties.

**Lemma 2.6** ((Chan et al., 2011; Wainwright, 2019)). *Let $\{X_i\}$ be a collection of independent random variables such that $X_i \sim \mathrm{Lap}(b_i)$ for all $1 \leq i \leq m$. Then, for $\nu \geq \sqrt{\sum_i b_i^2}$ and $0 < \lambda < \frac{2\sqrt{2}\nu^2}{b}$ for $b = \max_i\{b_i\}$, $\Pr[|\sum_i X_i| \geq \lambda] \leq 2 \cdot \exp(-\frac{\lambda^2}{8\nu^2})$. Furthermore, if $b = b_i$ for any $i \in [m]$ and $m \geq \log \beta$, we have $\Pr\left[|\sum_i X_i| \geq 2\sqrt{2} \cdot b\sqrt{m \log \beta}\right] \leq \frac{2}{\beta}$*

The Laplace mechanism is a commonly used class of differential privacy mechanisms.

**Definition 2.7** (Laplace mechanism (Dwork et al., 2006)). *For any function $f : \mathcal{X} \to \mathbb{R}^k$, the Laplace mechanism on input $x \in \mathcal{X}$ samples $\mathcal{Y}_1, \ldots, \mathcal{Y}_k$ independently from Lap($\frac{\mathrm{GS}_f}{\varepsilon}$) and outputs $M(x) = f(x) + (\mathcal{Y}_1, \ldots, \mathcal{Y}_k)$. The Laplace mechanism is $\varepsilon$-DP.*

# 3 DP RANGE SUBGRAPH COUNTING

We now present a differential privacy algorithm for range subgraph counting and provide a proof of its privacy and utility guarantees. Due to space constraints, we will focus on the algorithm and analysis for the one-dimensional case ($d = 1$) in this section, while the general case for $d \geq 2$ will be addressed in Appendix D.

## 3.1 THE ALGORITHM

**Overview of the algorithm and some definitions** Our algorithm for the case $d = 1$ consists of three steps:

(1) Map all the vertices in the input graph $G$ to points in a two-dimensional Euclidean space, where each point corresponds to a rank pair, which is a point $(a, b) \in [n]^2$ such that $a$ and $b$ represent the ranks of some vertices based on their attribute value and index order (see Algorithm 1). We construct a weight vector $\mathbf{w}$ for these points, with the weight of each point representing the number of occurrences that are "registered" at the corresponding rank pair (see PROJ($G, H$) in Algorithm 1).

(2) Build a range tree on the mapped points and the weight vector $\mathbf{w}$ such that each leaf node contains the weight corresponding to its point, while each internal node contains the sum of the weights of its children and bound information. Then, add Laplace noise to the weight of each node in the tree (see TREECONST($\mathbf{w}, \varepsilon, \mathrm{GS}_{f_H}$) in Algorithm 2).

(3) For any specified query $q$, traverse the tree to find the corresponding nodes and report their associated weights (see QUERY($G, H, Q, \varepsilon$) in Algorithm 3).

Here we make some additional symbol declarations. Recall that $V = [n]$. We use $u$ to represent the initial label of a vertex and use $s(u)$ to represent the *rank* a vertex after the second step of PROJ. Note that by definition, the ranks assigned to each vertex are unique.

**Definition 3.1.** *We say an occurrence of $H$ is* registered *at the vertex pair $(u, v)$ if $u, v \in V_H$ and $s(u) < s(u_1) < \cdots < s(u_{|V_H|-2}) < s(v)$.*

Note that for any occurrence of pattern graph $H$, it is mapped to a *unique* vertex pair $(u, v)$.

**Definition 3.2** (Discretization). *For any range query $q = [\ell, r]$, where $\ell, r \in \mathbb{R}$, we associate it with two vertices $u_\ell$ and $u_r$, where the attribute value of $u_\ell$ is the first one that is at least $\ell$, and the attribute value of $u_r$ is the last one that is at most $r$. In cases of ties, we select vertices based on the smallest lexicographical order.*

We note that even though the attributes are real values, we can discretize the problem as follows. The above discretization leads to the following useful fact:

**Fact 3.3.** *For all $Q$, the number of distinct subgraphs $G[V_q]$ induced by the queries in $Q$ is $O(n^2)$.*

For any range query $q = [\ell, r]$, we first apply the discretization described above to obtain a new range $q' = [s(u_\ell), s(u_r)]$. Note that the ranges $q$ and $q'$ correspond to the same subgraph. For simplicity, we will use $q = [\ell, r]$ to refer to the range corresponding to its discretized counterpart in the following. Now we describe our algorithm in more detail.

**Subgraph Counting Projection** The Algorithm 1 takes as input an $n$-vertex graph $G = (V, E, \mathbf{a})$, where each vertex has an associated attribute. First, it reorders the vertices based on their attribute values in ascending order, breaking ties by the initial vertex labels. For each occurrence of a subgraph $H$ in $G$, it updates a weight vector $\mathbf{w}$, where each element corresponds to a vertex pair, and the weight reflects the number of subgraph occurrences involving that pair. The algorithm then returns the weight vector $\mathbf{w}$, representing the counts of subgraph occurrences for all vertex pairs.

---

**Algorithm 1** PROJ($G = (V, E, \mathbf{a}), H$)    ▷ Subgraph Counting Projection

1: **Input**: An $n$-vertex graph $G = (V, E, \mathbf{a})$.
2: Sort vertices by attribute value in ascending order. For vertices with the same attribute value, sort by their initial labels. Let $s : V \to [n]$ denote the rank.
3: Initialize $w_{(u,v)} = 0$, for all $u, v \in V$.
4: **for all** occurrences of subgraph $H$ in $G$ **do**
5:    Compute $w_{(s(u),s(v))} = w_{(s(u),s(v))} + 1$, where the occurrence is registered at $(u, v)$.
6: **end for**
7: **return** $\mathbf{w} = \{w_{(s(u),s(v))}\}$

---

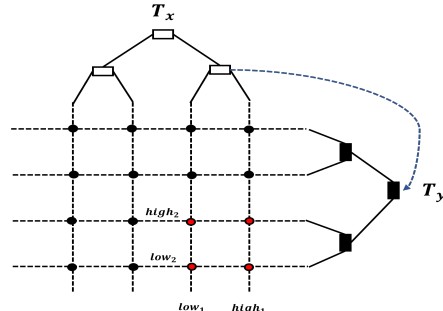

Figure 1: Schematic diagram of the 2D Range Tree used in our work. The detail can be seen in Appendix B.

**DP Range Tree Construction.** In Algorithm 2, we map all vertex pairs to points on a 2D plane based on their ranks $s(u)$ and $s(v)$, where each point has an associated weight $w_{(s(u),s(v))}$, representing the subgraph occurrences involving the corresponding vertex pair. We then utilize a range tree to preprocess these points and efficiently answer range subgraph counting queries.

A range tree is a binary tree structure designed for interval queries and summation. It recursively decomposes the interval and precomputes the sums at each node, where each node stores interval boundaries and corresponding sum values. In the TREECONST algorithm (Algorithm 2), we implement a modified version of a 2D range tree, which is fully described in Appendix B. This enables efficient querying for subgraph counts within specified ranges while preserving differential privacy.

The schematic diagram of the 2D range tree is shown in Figure 1. Intuitively, the tree construction process recursively divides the $n \times n$ points on the 2D plane into two equal parts, with each tree node storing interval boundaries and corresponding weight sums.

We begin by partitioning the first dimension of the plane to construct a tree $T_x$, where each node in $T_x$ corresponds to a sub-interval of this dimension. For each node in $T_x$, we then partition the second dimension to construct a one-dimensional range tree $T_y$. As a result, each node in $T_x$ represents a range tree $T_y$, and each node in $T_y$ corresponds to a sub-interval within the 2D space. Finally, to ensure differential privacy, Laplace noise is added to the weight of each node in $T_y$ (not both $T_y$ and $T_x$ add noise).

**Algorithm 2** TREECONST($\mathbf{w}, \varepsilon, \mathrm{GS}_{f_H}$)
▷ Private Range Tree Construction

1: **Input:** Projection vector $\mathbf{w}$, privacy parameter $\varepsilon > 0$, and global sensitivity $\mathrm{GS}_{f_H}$.
2: Construct $T_x$ according to Definition B.3 using tuples $(s(u), s(v), w_{(s(u),s(v))})$, where $u, v \in V$.
3: Create a noisy version, $\widetilde{T}_x$, by adding Laplace noise to the weight of each node in every $T_y$ tree (within each node of $T_x$). Specifically, update the weight as weight $=$ weight $+ \mathrm{Lap}(\frac{t}{\varepsilon})$, where $t = \mathrm{GS}_{f_H} \cdot \log^2 n$.
4: **return** $\widetilde{T}_x$

**Algorithm 3** QUERY($G, H, Q, \varepsilon$)   ▷ Private Range Subgraph Counting Query

1: **Input**: An $n$-vertex graph $G = (V, E, \mathbf{a})$, a pattern graph $H$, a set of range queries $Q$, and privacy parameter $\varepsilon$.
2: Compute the global sensitivity: $\mathrm{GS}_{f_H} = f_H(K_n) - f_H(K_n - e)$.
3: Compute the projection vector: $\mathbf{w} = \mathrm{PROJ}(G, H)$.
4: Construct the noisy range tree: $\widetilde{T}_x = \mathrm{TREECONST}(\mathbf{w}, \varepsilon, \mathrm{GS}_{f_H})$.
5: **for** each query $q \in Q$ **do**
6:   Determine $\ell$ and $r$ according to Definition 3.2.
7:   **return** the result of Definition B.4 using $\widetilde{T}_x$ and the range $[\ell, n] \times [1, r]$.
8: **end for**

**Query procedure.** For each query $q$, we discretize the range $[\ell, r]$ and access the range tree $\widetilde{T}_x$ to obtain the result. Specifically, we need to calculate the sum of the weights of the selected nodes in $T_y$. In Algorithm 3, the process involves locating the relevant node in $T_x$ and subsequently identifying the corresponding nodes in $T_y$ by traversing from top to bottom (see Figure 1).

## 3.2 THE ANALYSIS

We will make use of the following fact.

**Fact 3.4** (Properties of Range Tree). *Each range tree is a binary tree with a depth of $\log n$. Each leaf node stores the interval bounds and the sum value, along with the root node of the nested tree. The sum of the values of the tree nodes equals the sum of the values of the left child plus the sum of the values of the right child.*

**Privacy**   We now prove that Algorithm 3 is an $\varepsilon$-DP algorithm.

**Lemma 3.5.** *Assuming the weight $w_{(s(u),s(v))}$ of each pair is generated by Algorithm 1, the number of occurrences of $H$ in the graph, consisting of all vertices within the range $q = [\ell, r]$, is equal to the sum of the weights of all rank pairs falling within the range $[\ell, n] \times [1, r]$. That is, $f_H(G_q) = \sum_{(s(u),s(v)) \in [\ell,n] \times [1,r]} w_{(s(u),s(v))}$.*

*In particular, the number $f_H(G)$ of pattern graphs $H$ in $G$ is equal to $\sum_{(u,v) \in V \times V} w_{(u,v)}$.*

*Proof.* If an occurrence of $H$ falls within the range $q = [\ell, r]$, it means that all vertices in this occurrence of $H$ are contained within the range $q$. Specifically, if an occurrence of $H$ is registered at the vertex pair $(u, v)$, then the ranks satisfy $\ell \leq s(u) < s(u_1) < \cdots < s(u_{|V_H|-2}) < s(v) \leq r$.

Since the vertex reordering is performed in the second step of Algorithm 1 and each vertex is assigned a unique rank, we can transform the subgraph range into a range on the weight vector $\mathbf{w}$. Consequently, the sum of the weights of all rank pairs in the range $[\ell, n] \times [1, r]$ corresponds to the number of occurrences of subgraph $H$ that fall within the range $q = [\ell, r]$. $\square$

**Lemma 3.6.** *Algorithm 3 is $\varepsilon$-DP.*

*Proof.* We use $\mathbf{w}$ and $\mathbf{w}'$ to denote the different weight vectors formed by graphs $G$ and $G'$, respectively, where $G \sim G'$ (i.e., $G$ and $G'$ differ by a single edge). The global sensitivity of function $f$ is denoted as $\mathrm{GS}_f$, and the global sensitivity of $f_H$ is defined as $\mathrm{GS}_{f_H} = \max_{G \sim G'} |f_H(G) - f_H(G')|$ (see Definition 2.1). The sensitivity of $\mathbf{w}$, denoted as $\mathrm{GS}_{\mathbf{w}}$, is defined as $\max_{\mathbf{w}, \mathbf{w}'} \|\mathbf{w} - \mathbf{w}'\|_1$. Note that for any $\mathbf{w}, \mathbf{w}'$, we have $\|\mathbf{w} - \mathbf{w}'\|_1 = |\|\mathbf{w}\|_1 - \|\mathbf{w}'\|_1|$. This follows from the fact that subgraph counting is a monotonic function, meaning that the addition of any edge does not reduce the number of occurrences of $H$. Furthermore, each element of $\mathbf{w}$ or $\mathbf{w}'$ is non-negative.

Thus, the global sensitivity of $\mathbf{w}$ is $\text{GS}_{\mathbf{w}} = \max_{\mathbf{w},\mathbf{w}'} \|\mathbf{w} - \mathbf{w}'\|_1 = \max_{\mathbf{w},\mathbf{w}'} |\|\mathbf{w}\|_1 - \|\mathbf{w}'\|_1| = \max \left| \sum_{(u,v)\in V\times V} w_{(u,v)} - \sum_{(u,v)\in V\times V} w'_{(u,v)} \right| = \max_{G\sim G'} |f_H(G) - f_H(G')| = \text{GS}_{f_H}$, where the second to last equation follows from Lemma 3.5.

Let's revisit the algorithm with a focus on a layer of the range tree $T_x$. At each layer of $T_x$, we select all corresponding $T_y$ trees. The number of $T_y$ trees at this layer is equal to the number of nodes at that layer of $T_x$. For each layer $i$ (where $i \in [\log n]$), let $p$ represent all the nodes on the $i$-th layer of these $T_y$ trees. The sum of the weights of the selected nodes, $\sum_p p.\text{weight}$, equals the sum of the weights of all vertex pairs, which can be written as $\sum_{(u,v)\in V\times V} w_{(u,v)}$ (or equivalently, $\sum_{(u,v)\in V\times V} w_{(s(u),s(v))}$).

In other words, if we treat the weights obtained in this way as a vector, its sensitivity is equal to the sensitivity of $\mathbf{w}$, denoted by $\text{GS}_{\mathbf{w}}$.

Since $T_x$ has at most $\log n$ layers and each $T_y$ tree also has at most $\log n$ layers (as stated in Fact 3.4), there are at most $\log^2 n$ such vectors. Let $\mathbf{w}_t$ be the vector of weights from all nodes on the $T_y$ trees. The sensitivity of $\mathbf{w}_t$ is $GS_{\mathbf{w}_t} = \text{GS}_{\mathbf{w}} \cdot \log^2 n = \text{GS}_{f_H} \cdot \log^2 n$.

Thus, according to the Laplace mechanism and basic composition theorems, adding Laplace noise with magnitude $\text{GS}_{f_H} \cdot \log^2 n/\varepsilon$ to each element of the vector ensures that $T_x$ achieves differential privacy. For each query, the range trees $T_x$ and $T_y$ are reused, and hence Algorithm 3 maintains $\varepsilon$-differential privacy based on the post-processing property. $\square$

**Utility**  Now we analyze the utility of Algorithm 3. Interestingly, while the range tree approach is traditionally employed in non-private algorithms to improve query time, in this context, it also serves to reduce the errors introduced by differential privacy protection. By leveraging the structure of the range tree, we can distribute the noise more effectively across the tree's nodes, which minimizes the overall impact of noise on query accuracy. This ensures that the utility of the algorithm remains high, even with the added noise required to preserve privacy.

We first prove that for a query $q$, only a small number of noise terms are required to obtain the private answer. We have the following lemma whose proof is deferred to Appendix C.

**Lemma 3.7.** *For a given query $q$ and any pattern graph $H$, to calculate $f_H(G_q)$, the number of occurrence of $H$ in the graph $G_q$, we only need to sum the weights of at most $\log^2 n$ tree nodes.*

We will now show that the DP range subgraph counting implemented by our algorithm provides strong utility guarantees for $d = 1$, achieving an error that is close to that of DP global subgraph counting error (i.e., $\text{GS}_{f_H}$), differing only by a factor of $\log^C n$.

As outlined in Algorithm 2, we introduce Laplace noise to the weight of each node in the $T_y$ trees. Referring back to Lemma 3.7, we note that when answering a query, we only need to compute the sum of the weights of at most $\log^2 n$ nodes.

Assume that $p$ represents the $T_y$ nodes selected by query $q$, and each $Y_p$ is an independent random variable where $Y_p \sim \text{Lap}\left(\frac{\text{GS}_{f_H}\cdot\log^2 n}{\varepsilon}\right)$. Let $f_H(\cdot)$ denote the true result and $\widetilde{f}_H(\cdot)$ the output of the algorithm. For any query $q \in Q$, the additive error generated can be expressed as: $\left| f_H(G_q) - \widetilde{f}_H(G_q) \right| = \left| \sum_{p\in q} w(p) - \sum_{p\in q} \widetilde{w}(p) \right| \leq \left| \sum_p^{\log^2 n} Y_p \right| = O\left(\frac{\text{GS}_{f_H}\cdot\log^{3.5} n}{\varepsilon}\right)$, where the final inequality follows from the fact that each query utilizes at most $\log^2 n$ tree node weights for computation by Lemma 3.7. This bound holds with a probability of at least $1 - \frac{1}{n^3}$, as established by Lemma 2.6, where $b = \text{GS}_{f_H} \cdot \log^2 n$, $m = \log^2 n$, and $\beta = n^3$. We can derive the following bound: $\max_{q\in Q} \left| f_H(G_q) - \widetilde{f}_H(G_q) \right| = O\left(\frac{\text{GS}_{f_H}\cdot\log^{3.5}(n)}{\varepsilon}\right)$. This holds with a probability of at least $1 - \frac{1}{n}$. This result is achieved by applying the union bound, as there are at most $O(n^2)$ effective subgraphs by Fact 3.3. This finishes the proof for the case $d = 1$.

## 4 Experiments

To evaluate the trade-off between privacy and utility in our algorithm, We conducted experiments on two real-world datasets.

**Datasets: Ego-Facebook:** Facebook data was collected from survey participants using this Facebook app. The dataset includes node features (profiles), circles, and ego networks. The network (Leskovec & Mcauley, 2012) has $n = 4039$ and $m = 88234$.

**Fb-Pages-government:** Data collected about Facebook pages (November 2017). These datasets represent blue verified Facebook page networks of different categories. Nodes represent the pages and edges are mutual likes among them. The network (Leskovec & Mcauley, 2012) has $n = 7057$ and $m = 89455$. For each vertex in the aforementioned two networks, we sample values from a standard normal distribution to serve as vertex attributes.

**Infrastructure:** All algorithms are implemented in Python. We ran our experiments on a system with a 128-core Intel(R) Xeon(R) Platinum 8358 CPU @ 2.60GHz and 504GB RAM.

**Baseline:** There is no prior work on differential privacy range subgraph counting. We use two baselines for comparison. The first baseline **BASE_COMP** uses the Laplace mechanism and advanced composition theorem (Dwork et al., 2014), and we set $\delta = 0.01$. The second baseline **BASE_PRE** adds Laplace noise of size $\frac{\mathrm{GS}_{f_H}}{\varepsilon}$ on the basis of subgraph counting projection (PROJ) which is the same as Algorithm 1, and does not build a tree structure. We use **DPSRC** to represent our algorithm (pure-DP) and **DPSC** to represent global subgraph counting with privacy which only focus the whole graph and answer one query. We give the theoretical information of the above algorithm in Table 1. In our experiments, we set the attribute dimension $d = 1$.

**Metric:** We define the relative error for a query $q$ as $\frac{\left|\widetilde{f}_H(G_q) - f_H(G_q)\right|}{\min\left(f_H(G_q), 0.001n\right)}$. This metric follows the approach outlined in (Imola et al., 2021). To maintain a consistent standard, we ensure that all tested algorithms adhere to either $\varepsilon$-DP or $(\varepsilon, \delta)$-DP. We keep the query $q$ fixed and randomly generated across a series of experiments, ensuring that $|V_q| = \Theta(n)$. In fact, our algorithm can handle any number of queries, and compared to other algorithms, it demonstrates an advantage when the graph scale is larger, as shown in Theorem 1.

| Algorithm | Query Type | Privacy | Additive Error |
|---|---|---|---|
| BASE_COMP | Range | $(\varepsilon,\delta)$-DP | $\widetilde{O}(n \cdot \mathrm{GS}_{f_H})$ |
| BASE_PRE | Range | $\varepsilon$-DP | $\widetilde{O}(n \cdot \mathrm{GS}_{f_H})$ |
| DPSC | Global | $\varepsilon$-DP | $\widetilde{O}(\mathrm{GS}_{f_H})$ or instance-dependent[2] |
| **DPRSC** | **Range** | $\varepsilon$**-DP** | $\widetilde{O}(\mathbf{GS}_{f_H})$ |

Table 1: The performance guarantees of DP algorithms for counting occurrences of $H$. For range queries, the additive error is specified according to Theorem 1, while for single queries, it is measured by the absolute value of the difference between the algorithm's output and the actual count.

**Relative Error vs $\varepsilon$:** We evaluated the relation between relative error and $\varepsilon$. We tested the algorithm on the ego-facebook and fb-pages-government datasets for the cases when $H$ is triangle, 2-star and edge, respectively. Figure 2 describes the relationship between the relative error and $\varepsilon$ when the algorithm guarantees $\varepsilon$-DP ($(\varepsilon, \delta)$-DP) under the same random query. When $\varepsilon$ is relatively small, the privacy protection is strong, making it difficult for potential attackers to distinguish between any two inputs based on the output; however, the relative error is large. As $\varepsilon$ increases, privacy becomes weaker and the relative error becomes smaller. In addition, it can be seen that our algorithm is significantly better than the baseline. In practical applications, the choice of $\varepsilon$ should be made based on specific requirements.

**Relative Error vs $n$:** We evaluated the relation between relative error and $n$. We tested the algorithm on the ego-facebook and fb-pages-government datasets for the cases where $H$ is triangle, 2-star and edge, respectively under the same random query. In the experiment, we set $\varepsilon = 2.0$ and randomly generate a fixed query. As can be seen from Figure 3, our algorithm is significantly better than the baseline. And the experimental results are basically in line with intuition: the increase in graph size will lead to an increase in the number of triangles, 2-stars and edge in most cases. If the growth rate is greater than the growth rate of additive error, the relative error will decrease, and vice versa. Due

---

[2]The error is determined by certain unfixed properties of the input graph (such as the number of edges and the degree of the nodes). In the worst case, it is $\widetilde{O}(GS_{f_H})$, and the actual error may be smaller than usual.

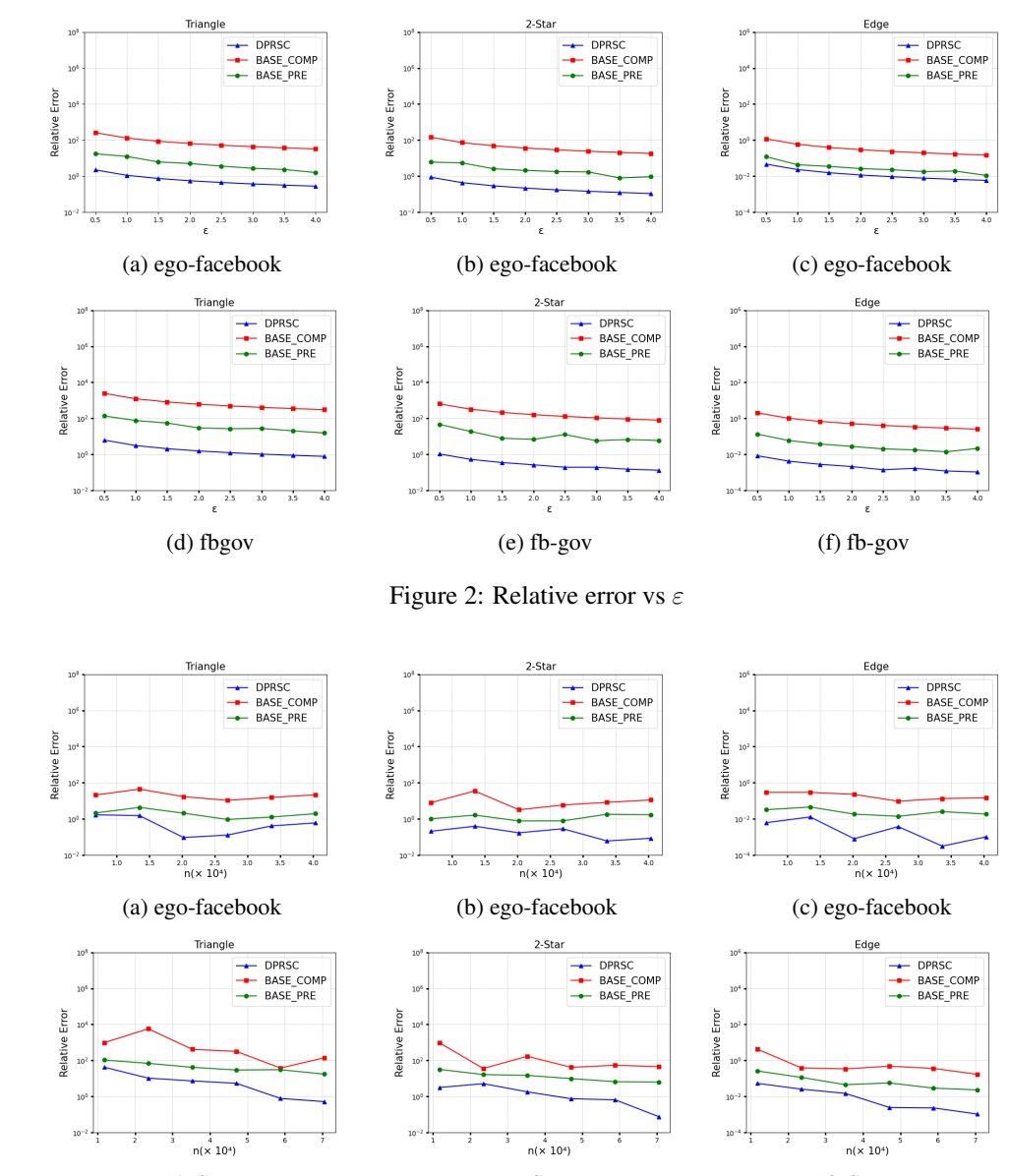

Figure 2: Relative error vs $\varepsilon$

Figure 3: Relative error vs $n$

to limitations in equipment and storage optimization, we believe that our algorithm demonstrates a more pronounced advantage on larger-scale graphs and queries, as the impact of the $\log n$ factor becomes less significant in such cases.

## 5 CONCLUSION

We give the first algorithm for the differentially private range subgraph counting problem that achieves nearly optimal additive error for any constant dimension $d$ and a constant privacy parameter $\varepsilon$. Our approach establishes a connection between subgraph counting and the range tree technique within the DP framework. Further exploration of instance-dependent error bounds for private range subgraph counting would be interesting. Another natural question is how to design an algorithm that ensures the additive error remains non-trivial, if the vertex attributes are high-dimensional (for example, $d = \Omega(\log n)$).

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

## A  UPPER BOUND ON THE GLOBAL SENSITIVITY OF SUBGRAPH COUNTING

In Section 3, we used $\mathrm{GS}_{f_H}$ to denote the global sensitivity of subgraph counting. In fact, in many cases, we do not know the exact value of $\mathrm{GS}_{f_H}$ or it is cumbersome to calculate, and we want to estimate it. Here we give an upper bound for $\mathrm{GS}_{f_H}$ through the *fractional edge-cover number*, an important metric in graph theory. We also demonstrate the existence of a pattern subgraph $H$ where $\mathrm{GS}_{f_H}$ meets the established upper bound. To the best of our knowledge, this work is the first to combine differential privacy for graphs with the concept of fractional edge-cover number.

| Pattern Graph | $\mathrm{GS}_{f_H}$ | $\rho(H)$ |
|---|---|---|
| Edge | $1$ | $1$ |
| Triangle | $n-2$ | $\frac{3}{2}$ |
| $k$-Star | $\binom{n-1}{k-2}$ | $k-1$ |
| $k$-Cycle | $(k-2)!\binom{n-2}{k-2}$ | $\frac{k}{2}$ |
| $k$-Clique | $\binom{n-2}{k-2}$ | $\frac{k}{2}$ |

Table 2: Global sensitivity $\mathrm{GS}_{f_H}$ and $\rho(H)$ of some common pattern graphs $H$

**Graph theory**  We introduce the definition of fractional edge-cover number in (Assadi et al., 2018) which is a classic definition of a subgraph enumeration and counting field.

**Definition A.1** (Fractional Edge-Cover Number). *A fractional edge-cover of $H(V_H, E_H)$ is a mapping $\phi : E_H \to [0,1]$ such that for each vertex $v \in V_H$, $\sum_{e \in E_H, v \in e} \phi(e) \geq 1$. The fractional edge-cover number $\rho(H)$ of $H$ is the minimum value of $\sum_{e \in E_H} \phi(e)$ among all fraction edge covers $\phi$.*

Atserias, Grohe, and Marx (Atserias et al., 2008) established a relationship between the number of occurrences of $H$ in a graph, the number of edges, and the fractional edge-cover number.

**Lemma A.2** ((Atserias et al., 2008)). *The number of occurrences of $H$ in a graph $G$ with $m$ edges is $O(m^{\rho(H)})$.*

This lemma states that for any graph $G$, if the number of edges in the graph is $m$, then the number of occurrences of subgraph $H$ in $G$ is $O(m^{\rho(H)})$. For example, if $H$ is a triangle, we can obtain $\rho(H) = \frac{3}{2}$ according to the definition of fractional edge-cover number. It means that the number of triangle in a graph is $O(m^{\frac{3}{2}})$, that is $O(n^3)$ when the graph is complete graph with $n$ vertices. It is known that one can efficiently compute the fractional edge cover $\rho(H)$ in polynomial (in $|H|$) time (see (Assadi et al., 2018)).

We try to bound $\mathrm{GS}_{f_H}$ in a simple and effective way. We need to understand the global sensitivity of the subgraph count in the graph, which is actually to calculate the number of occurrences of $H$ that contain a specific vertex pair $(i, j)$ in the complete graph.

**Lemma A.3.** *Given an $n$ vertex graph $G$, pattern graph $H$. The upper bound of $\mathrm{GS}_{f_H}$ is $O(n^{2\rho(H)-2})$.*

*Proof.* $\mathrm{GS}_{f_H}$ is global sensitivity of subgraph $H$ counting, note that

$$\mathrm{GS}_{f_H} = \max_{G \sim G'} |f_H(G) - f_H(G')| = |f_H(K_n) - f_H(K_n - \{(i,j)\})| = O(n^{2\rho(H)-2})$$

The second equality holds because the global sensitivity of $f_H$ is equal to the difference between the count of the complete graph $K_n$ and the count of the complete graph $K_n$ with one edge $(i, j)$ missing. The final equality follows from Lemma A.2. $\qquad\square$

## B  RANGE TREE IN ALGORITHM 2 AND ALGORITHM 5

For clarity, we define the tree construction and query process to streamline the algorithm's description. Here, $T$, $T_x$, and $T_y$ all represent trees. The construction of the range tree is based primarily

on (Bentley & Saxe, 1978), with minor modifications. A schematic of the 2D range tree is provided in Figure 1.

We begin by introducing the basic 1D range tree.

**Definition B.1** (1D Range Tree). *Given a set of points $P = \{(x_i, w_i)\}$, where each point has an $x$-coordinate and $weight$, the $1D$ range tree is constructed as follows:*

1. *Sort the points by $x$-coordinates, denoted as $x_1, \ldots, x_n$.*

2. *Begin building the tree recursively from the root node, where the interval spans from $x_1$ to $x_n$.*

3. *For a given interval $x_l, \ldots, x_r$ corresponding to a tree node $p$, set $mid = \frac{l+r}{2}$. Recursively construct the left child using points $x_l, \ldots, x_{mid}$ and the right child using points $x_{mid+1}, \ldots, x_r$. If the interval contains only one point, terminate the recursion.*

4. *During backtracking, compute the weight of the current tree node as the sum of its interval:*

$$node.weight = left.weight + right.weight.$$

**Definition B.2** (1D Range Tree Query). *Given a query range $[low, high]$, start at the root node of the 1D range tree $T$.*

1. *Start the recursive query from the root node of $T$.*

2. *For the current $node$, if node falls within the range $[low, high]$, return $p.weight$. If $low$ lies within the left child of $p$, recursively query the left subtree; if $high_1$ lies within the right child, recursively query the right subtree.*

3. *When backtracking, sum the results of the left child and the right child and return them.*

Next, we introduce a more complex case. To correspond to our chapter, we separate the 2D case and the $k$D ($k > 2$) case.

**Definition B.3** (2D Range Tree Construction). *For a set of points $P = \{(x_i, y_i, w_i)\}$ where each point has coordinates $(x, y)$ and $weight$, the 2D range tree is constructed as follow:*

1. *Group the points by their $x$-coordinates, and sort each group by $x$, denoted as $p_1, \ldots, p_n$.*

2. *Construct the 2D range tree $T_x$ using $p_1, \ldots, p_n$ in a similar approach to the 1D range tree, partitioning the first dimension. Note that each node of $T_x$ contains an associated 1D range tree $T_y$ for the second dimension.*

3. *For each node in $T_x$, take the points covered by that node, group them by their $y$-coordinates, sort them, and construct a corresponding range tree $T_y$ which is contained in the node $T_x$.*

**Definition B.4** (2D Range Tree Query). *Given $[low_1, high_1] \times [low_2, high_2]$ and a 2D Range Tree $T_x$, the query process is as follows:*

1. *Start the recursive query from the root node of $T_x$.*

2. *For the current $node$, if node falls within the range $[low_1, high_1]$, perform a query on $T_y$ with $[low_2, high_2]$ (call 1D tree query). If $low_1$ lies within the left child of $node$, recursively query the left subtree; if $high_1$ lies within the right child, recursively query the right subtree.*

3. *When backtracking, sum the results of the left child and the right child and return them.*

Next we describe range tree construction and query in general.

**Definition B.5** ($k$D Range Tree Construction). *For a set of points $P = \{(x_i^1, \ldots, x_i^k, w_i)\}$ where each point has coordinates $(x^1, \ldots, x^k)$ and $weight$, the $k$D range tree is constructed as follow:*

1. *Group the points by their first dimension, and sort each group by the first dimension, denoted as $p_1, \ldots, p_n$.*

2. *Construct the kD range tree $T_1$ using $p_1, \ldots, p_n$ in a similar approach to the 1D range tree, partitioning the first dimension. Note that each node of $T_1$ contains an associated $(k-1)D$ range tree $T_2$ for the second dimension, recursively.*

3. *For each node in $T_1$, take the points covered by that node, group them by their second dimension, sort them, and construct a corresponding $(k-1)D$ range tree $T_2$.*

**Definition B.6** (*kD Range Tree Query*). *Given a k-dimensional query range $[low_1, high_1] \times \cdots \times [low_k, high_k]$ and kD range tree $T$:*

1. *Start the recursive query from the root node of $T_1$.*

2. *For the current $node$, if $node$ falls within the range $[low_1, high_1]$, perform a query on $T_2$ with $[low_2, high_2]$ (call $(k-1)D$ tree query recursively). If $low_1$ lies within the left child of $node$, recursively query the left subtree; if $high_1$ lies within the right child, recursively query the right subtree.*

3. *When backtracking, sum the results of the left child and the right child and return them.*

## C  PROOF OF LEMMA 3.7

Given a query $q = [\ell, r]$, we can prove that only at most $\log^2 n$ tree node weights of $T_y$ are needed to compute the result.

First, consider the tree $T_x$, which represents the first dimension (the rank of vertex pairs based on their first vertex). Our task is to select the tree nodes that cover the range $[\ell, n]$. In the binary range tree structure, once a parent node is selected, its child nodes are not selected since the parent already covers the required range. This simplifies the problem to identifying nodes whose first dimension (rank) is numbered in $i, i+1, \ldots, n$.

At the $i$-th level (from bottom to top, i.e., levels $1, 2, \ldots, \log n$), each tree node at this level covers intervals such as $[1, 2^i], [2^i + 1, 2^{i+1}], \ldots, [2^{\log n - 1} + 1, n]$.

Assume $j$ is the smallest rank not less than $\ell$. We can represent the difference $n - j$ as a binary number, which can be expressed as a sum of at most $\log(n-j)$ powers of 2. For example, the number 10 in binary is 1010, i.e., $10 = 2^3 + 2^1$. Similarly, we can cover the range $[\ell, n]$ by selecting at most $\log n$ nodes in $T_x$, since the range tree is built based on binary subdivisions of the range.

Similarly, for each node in $T_x$ that we select, it contains a nested tree $T_y$. At this stage, for each $T_y$, we select a tree node corresponding to the range $[1, r]$ (since we have already determined the left boundary). Just like before, we can cover all rank pairs whose second dimension is in $[1, r]$ by selecting at most $\log n$ tree nodes from $T_y$.

Thus, by selecting the necessary nodes in both $T_x$ and $T_y$, we can cover all rank pairs falling within $[\ell, n] \times [1, r]$. This allows us to retrieve all subgraph counts where the vertices lie in the range $[\ell, r]$.

In summary, we need to select at most $\log^2 n$ tree nodes from $T_y$ to find all rank pairs within $[\ell, n] \times [1, r]$. According to Lemma 3.5, the number of subgraphs with vertex ranks falling within any given query range can be efficiently calculated.

Note that we ignore some rounding issues here.

## D  MISSING ALGORITHM AND PROOF OF THEOREM 1: THE CASE $d \geq 2$

In the previous section we discussed the case of one-dimensional attribute for a vertex. In this section, we extend our algorithm to the case of multi-dimensional (low-dimensional) attribute for a vertex which is a more general situation, i.e. $\mathbf{a}(u) \in \mathbb{R}^d$, where $u \in V$.

Without loss of generality, we assume that each attribute $\mathbf{a}_i(u) \in [0, \lambda_i]$ for $i \in [d]$, and each query $q = [l_1, h_1] \times \cdots \times [l_d, h_d]$.

When vertex attributes are multi-dimensional, the algorithm needs some adjustments. The entire algorithm PROJMULT, TREECONSTMULT and QUERYMULT is given in this section.

---

**Algorithm 4** PROJMULT ($G = (V, E, \mathbf{a}), H$)     ▷ Subgraph Counting Projection For Mult-attribute

---

1: **Input:** An $n$-vertex graph $G = (V, E, \mathbf{a})$.
2: Reorder all vertex labels by $i$-th attribute value from small to large. If the attribute values are the same, sort according to the initial label. Obtain the new rank $s_i : V \to [n]$ where $i \in [d]$.
3: Initialize $w_{(u_1, v_1, \ldots, u_d, v_d)} = 0$, for any $u_1, \ldots, u_d \in V$.
4: **for all** occurrences of subgraph $H$ in $G$ **do**
5:      Compute $w_{(s_1(u_1), s_1(v_1), \ldots, s_2(u_d), s_2(v_d))} = w_{(s(u_1), s(v_1), \ldots, s_d(u_d), s_d(v_d))} + 1$, where $u_i$ (resp. $v_i$) be the vertex in this occurrence with the smallest (resp. largest) rank in dimension $i$.
6: **end for**
7: **return** $\mathbf{w} = \{w_{(s(u_1), s(v_1), \ldots, s(u_d), s(v_d))}\}$

---

**Algorithm 5** TREECONSTMULT ($\mathbf{w}, \varepsilon, \text{GS}_{f_H}$)     ▷ Private Range Tree Contruction For Mult-attribute

---

1: **Input:** Projection $\mathbf{w}$, privacy parameter $\varepsilon > 0$ and global sensitivity $\text{GS}_{f_H}$.
2: Create a noisy version, $\widetilde{T}_1$, by adding Laplace noise to the weight of each node in every $T_d$ tree (within each node of $T_x$). Specifically, update the weight as weight = weight + $\text{Lap}(\frac{t}{\varepsilon})$, where $t = \text{GS}_{f_H} \cdot \log^{2d} n$.
3: **return** $\widetilde{T}_1$

---

**Algorithm 6** DPRSC ($G, H, Q, \varepsilon$)     ▷ Private Range Subgraph Counting Query For Mult-attribute

---

1: **Input:** An $n$-vertex graph $G = (V, E, \mathbf{a})$, a pattern graph $H$, a set of range queries $Q$, and privacy parameter $\varepsilon$.
2: $\text{GS}_{f_H} = f_H(K_n) - f_H(K_n - e)$.
3: $\mathbf{w} = \text{PROJMULT}(G, H)$.
4: $\widetilde{T}_1 = \text{TREECONSTMULT}(\mathbf{w}, \varepsilon, \text{GS}_{f_H})$
5: **for** $q \in Q$ **do**
6:      Get $\ell_i$ and $r_i$ according to Definition 3.2 for each dimension of $q$.
7:      **return** Output of Definition B.6 with $\widetilde{T}_1$ and $[\ell_1, r_1] \times \cdots \times [\ell_d, r_d]$.
8: **end for**

---

We refer to Definition 3.2 for the discretization steps in each dimension. Also, we abuse $\ell_i, r_i$ to denote rank for range.

**Fact D.1.** *For all $Q$, we have $|\{G[V_q] \mid q \in Q\}| = O(n^{2d})$.*

We say the vertex $u$ falls within query $q$ if $\mathbf{a}(u)$ satisfy $\mathbf{a}_i(u) \in [l_i, h_i]$ for $i \in [d]$. If we say that the vertices $(u_1, u_2, \ldots, u_k)$ *falls within* the range $q$ if and only if all vertices within the tuple fall within the range.

Inspired by the case where $d = 1$, we can still perform subgraph counting projection on the vertices of the graph. However, instead of projecting onto a plane, we project onto a hyperrectangle. Each range subgraph counting query actually queries a small hyperrectangle inside the large hyperrectangle and calculates the sum of the weights of the tuple in the small hyperrectangle. Similar to Section 3, we construct a nested tree based on these projections, ensuring that the tree with the finest granularity has noisy weights. The final result of each query is still determined by the node weights within the trees.

For the private range subgraph counting algorithm with multi-dimensional attributes, we give an algorithm with performance guarantee given in Theorem 1 and prove its privacy and utility.

### D.1 PROOF OF THEOREM 1

**Lemma D.2.** *Assuming that the weight $w_{(s_1(u_1), s_1(v_1), \ldots, s_d(u_d), s_d(v_d))}$ of each pair is generated by Algorithm 1, the number of occurrences of $H$ in the graph consisting of all vertices falling within*

the range $q = [\ell, r]$ is equal to the sum of the weights of all rank pairs falling within the range $[\ell, n] \times [1, r]$. That is,

$$f_H(G_q) = \sum_{(s_1(u_1),s_1(v_1),\ldots,s_d(u_d),s_d(v_d)) \in [\ell_1,n] \times [1,r_1] \times \cdots \times [\ell_d,n] \times [1,r_d]} w_{(s_1(u_1),s_1(v_1),\ldots,s_d(u_d),s_d(v_d))}.$$

In particular, the number $f_H(G)$ of pattern graphs $H$ in $G$ is equal to $\sum_{(u_1,v_1,\ldots,u_d,v_d) \in V^d} w_{(u_1,v_1,\ldots,u_d,v_d)}$ where $V^d = \underbrace{V \times V \times \cdots \times V}_{d}$.

*Proof.* First, We use tuples of length $2d$ to register an occurrence of the pattern graph $H$. Assume that $\mathbf{a}(u) = (\mathbf{a}_1(u), \ldots, \mathbf{a}_d(u))$, we construct rank tuple $(s_1(u_1), s_1(v_1), \ldots, s_d(u_{2d-1}), s_d(v_{2d}))$ to register subgraph $H$.

We say if an occurrence of pattern graph $H$ falls within range $q$, tuple must fall in query. In Algorithm 4, $d$ new sort $s$ is generated, we call the ordering of each dimension $s_i$. We suppose an occurrence can be registered at $(u_1, v_1, \ldots, u_d, v_d)$. If an occurrence of pattern graph $H$ falls within range $q$, that means

$$\ell_i \leq s_i(u) < s_i(u_1) < \cdots < s_i(u_{|V_H|-2}) < s_i(v) \leq r_i$$

for $i \in [d]$ and $u \in V$. Note that we have discretized the query $q$ similar to Definition 3.2, so $l_i$ and $r_i$ is discretized into rank.

Because of rearrange, each vertex has a unique sorting number, so each occurrence is registered at unique tuple. When all vertices in the tuple are within the range of $q$, all vertices in all subgraphs represented by the tuple also fall within this range. According to this corresponding relationship, we can obtain the sum of the weights of the tuples falling into $[l_1, n] \times [1, r_1] \times \cdots \times [l_d, n] \times [1, r_d]$ is equivalent to the number of occurrences of $H$ in the subgraph consisting of vertices in $[l_1, r_1] \times \cdots \times [l_d, r_d]$. $\square$

**Lemma D.3.** *Algorithm 6 is $\varepsilon$-DP.*

*Proof.* The proof method is an extension of Lemma 3.6. $\mathrm{GS}_\mathbf{w} = \mathrm{GS}_{f_H}$. And here $T_i$ has $\log n$ layers by Fact 3.4 for $i \in [d]$. Note that our approach in Section 3 can be extended to the case $d \geq 2$. If we combine the node weights of all $T_d$ into a vector, then this vector $\mathbf{w}_t$, then the global sensitivity of this vector is $\mathrm{GS}_{\mathbf{w}_t} = \mathrm{GS}_\mathbf{w} \cdot \log^{2d} n = \mathrm{GS}_{f_H} \cdot \log^{2d} n$. And there are $\log^{2d-1} n$ groups of $T_d$ that can form the entire point (tuple) set, that is $\mathrm{GS}_{f_H} \cdot \log^{2d} n$. $\square$

**Lemma D.4.** *For a given query $q$ and any pattern graph $H$, to calculate $f_H(G_q)$, the number of occurrence of $H$ in the graph $G_q$ induced by all vertices within the range, we only need to sum the weights of at most $\log^{2d} n$ tree nodes. In particular, the theorem degenerates into Lemma 3.7 when $d = 1$.*

*Proof.* Given a query $q = [l_1, r_1] \times \cdots \times [l_d, r_d]$, we can prove that only at most $\log^{2d} n$ tree node weights of $T_d$ are needed. In Lemma 3.7, we proved the case where $d = 1$. We use mathematical induction to prove the case where $d \geq 2$.

First, assume that when the dimension is $j - 1$ only the weight of $\log^{2j-2} n$ tree nodes is required.

We focus on the $T_{2j-1}$. Note that we need to find tree nodes that fall within $[l_j, n]$ from top to bottom. And once a parent node is selected, its children will not be selected. We can simplify the problem to selecting rank tuple whose $2j - 1$-th dimension points are numbered in $i, i+1, \ldots, n$. At the $i$-th level, each tree node in this level is responsible for interval numbers $[1, 2^i], [2^i + 1, 2^{i+1}], \ldots, [2^{\log n-1} + 1, n]$. In a similar way to Lemma 3.7, $\log n$ nodes in $T_{2j-1}$ is needed.

Similarly, each node in the $T_{2j-1}$ we select contains a $T_{2j}$. At this time, for each $T_{2j}$, select a tree node in the range $[1, r_j]$ (we have already determined the left boundary). Similarly, we can cover all rank tuple which $2j$-th dimension is in $[1, r_j]$ by selecting at most $\log n$ tree nodes. Then we can obtain all rank pair in range $[l_j, n] \times [1, r_j]$ and obtain all subgraph counting which vertex in $[l_j, r_j]$.

Recall that for the first $j-1$ dimensions, each query requires visiting $\log^{2j-2} n$ tree nodes. On this basis, to continue covering the remaining query dimension $[l_j, r_j]$ requires $\log n$ nodes. Therefore, for $j$ dimensions, each query requires $(\log^{2j-2} n) \cdot (\log^2 n) = \log^{2j} n$ nodes. Let $j = d$, we finish the proof. $\qquad\square$

Assume that $p$ represents the $T_d$ nodes selected by query $q$ and each $Y_p$ are independent random variables, where $Y_p \sim \text{Lap}(\frac{\text{GS}_{f_H} \cdot \log^{2d} n}{\varepsilon})$. For a fixed query $q$, the additive error generated is

$$\left| f_H(G_q) - \widetilde{f}_H(G_q) \right| = \left| \sum_p w(p) - \sum_p \widetilde{w}(p) \right| \le \left| \sum_p^{\log^{2d} n} Y_p \right| = O(\frac{\text{GS}_{f_H} \cdot d \cdot \log^{3d+0.5} n}{\varepsilon})$$

with a probability of at least $1 - \frac{1}{n^3}$ by Lemma 2.6 which $b = \text{GS}_{f_H} \cdot \log^2 n$, $m = \log^{2d} n$ and $\beta = n^{2d+1}$. by Lemma D.2, Lemma D.4, and Lemma 2.6.

# E   MISSING PROOF OF THEOREM 2

(Nguyen et al., 2023) introduced the concept of higher order local sensitivity to generalize to the DP general subgraph counting problem. Since directly adding noise to the local sensitivity can lead to privacy leakage, their approach is to estimate the noisy local sensitivity. If the local sensitivity of the local sensitivity still results in privacy leakage, further noise estimation is required for the local sensitivity of the local sensitivity, and this process is repeated recursively. We leverage their work to assist in the proof.

First, we introduce the concept of local sensitivity. The *local sensitivity* of $f$ is defined as

$$\text{LS}_f(G) = \max_{G':G' \sim G} \|f(G) - f(G')\|_1.$$

Let $S$ be a set of vertex pairs. Let $f_H(G, S)$ denote the number of occurrences of a fixed pattern graph $H$ in the graph $(V(G), E(G) \cup S)$. We define

$$f_H^{(k)}(G) = \max_{|S|=k} f_H(G, S).$$

We denote the output of Algorithm 7 as $\widetilde{\text{HS}}_{f_H}^{(k)}(G)$. Specifically, the noisy estimate of local sensitivity $\widetilde{\text{LS}}_{f_H}(G)$ is equivalent to $\widetilde{\text{HS}}_{f_H}^{(1)}(G)$. For clarity, we refer to $\widetilde{\text{HS}}_{f_H}^{(1)}(G)$ as $\widetilde{\text{HS}}_{f_H}(G)$.

---

**Algorithm 7** ESTIMATEHS$(G, H, \varepsilon', \delta')$ $\qquad \triangleright$ Estimating higher-order private local sensitivity (Nguyen et al., 2023), Algorithm 5

1: **Input**: An $n$-vertex graph $G$, privacy parameters $\varepsilon' > 0$ and $0 < \delta' < 1$.
2: Let $k_H = |E_H|$, $\widetilde{\text{HS}}_{f_H}^{(k_H)} = 0$.
3: **for** $k = k_H - 1$ down to 1 **do**
4: $\quad \widetilde{\text{HS}}_{f_H}^{(k)}(G) = f_H^{(k)}(G) + \widetilde{\text{HS}}_{f_H}^{(k+1)}(G)\frac{\ln 1/\delta'}{\varepsilon'} + \text{Lap}(\widetilde{\text{HS}}_{f_H}^{(k+1)}(G)/\varepsilon')$
5: **end for**
6: **return** $\widetilde{\text{HS}}_{f_H}(G)$

---

The following lemmas were proven in (Nguyen et al., 2023).

**Lemma E.1** ((Nguyen et al., 2023)). *Let* $\widetilde{\text{HS}}_{f_H}^{(k)}(G) = f_H^k(G) + \widetilde{\text{HS}}_{f_H}^{(k+1)}(G)\frac{\ln 1/\delta'}{\varepsilon'} + \text{Lap}(\widetilde{\text{HS}}^{(k+1)}(G)/\varepsilon')$, *for* $k = |E_H| - 1, \dots, 1$ *as computed in Algorithm 7. Then* $\widetilde{\text{HS}}_{f_H}(G)$ *is a* $(k_H\varepsilon', \delta' + k_H e^{\varepsilon'}\delta')$-DP *estimate of local sensitivity.*

**Lemma E.2** ((Nguyen et al., 2023)). *It holds that*

$$\Pr[\widetilde{\mathrm{HS}}_{f_H}^{(k)}(G) \geq f_H^{(k)}(G)] \geq 1 - \delta'$$

*for $k = 1, \ldots, k_H - 1$.*

The proof of Lemma E.3 follows the proof of Lemma 4.4 in (Karwa et al., 2011), and we extend their result to the case of multi-dimensional function $f$.

**Lemma E.3.** *Let $d \geq 1$. Let $\mathcal{B}$ be an $(\varepsilon_1, \delta_1)$-DP algorithm such that $\Pr[\mathcal{B}(x) \geq \mathrm{LS}_f(x)] > 1 - \delta_2$ for all $x$. Consider the algorithm $\mathcal{A}$ that runs $\mathcal{B}(x)$ to obtain an estimate $\widetilde{\mathrm{LS}}_x$ of the local sensitivity, and releases both $\widetilde{\mathrm{LS}}_x$ and a noisy estimate of $f$, i.e.,*

$$\mathcal{A}(x) = (\widetilde{\mathrm{LS}}_x, f(x) + \mathrm{Lap}^d(\widetilde{\mathrm{LS}}_x/\varepsilon_2)),$$

*where $\widetilde{\mathrm{LS}} = \mathcal{B}(x)$, $\mathrm{Lap}^d(b)$ represents a $d$-dimensional vector such that each element is independently sampled from a Laplace distribution with mean $0$ and scale parameter $b$. Then $\mathcal{A}$ is $(\varepsilon_1 + \varepsilon_2, \delta_1 + e^{\varepsilon_1}\delta_2)$-DP.*

*Proof.* Given neighboring datasets $x$ and $x'$, where $f(x), f(x') \in \mathbb{R}^d$, consider the following:

$$\mathcal{A}(x) = (\widetilde{\mathrm{LS}}_x, f(x) + \mathrm{Lap}^d(\widetilde{\mathrm{LS}}_x/\varepsilon_2))$$

$$\mathcal{A}(x') = (\widetilde{\mathrm{LS}}_{x'}, f(x') + \mathrm{Lap}^d(\widetilde{\mathrm{LS}}_{x'}/\varepsilon_2))$$

where $\widetilde{\mathrm{LS}}_x = \mathcal{B}(x)$ and $\widetilde{\mathrm{LS}}_{x'} = \mathcal{B}(x')$. Now, define the random variable

$$\mathcal{A}_{\mathrm{mix}} = (\widetilde{\mathrm{LS}}_x, f(x') + \mathrm{Lap}^d(\widetilde{\mathrm{LS}}_x/\varepsilon_2)).$$

Let $p_x$, $p_{x'}$ and $p_{\mathrm{mix}}$ be the probability distributions of $\mathcal{A}(x)$, $\mathcal{A}(x')$ and $\mathcal{A}_{\mathrm{mix}}$. First, consider the difference between $\mathcal{A}(x')$ and $\mathcal{A}_{\mathrm{mix}}$. They differ only in the initial estimate $\widetilde{\mathrm{LS}}$ (either $\mathcal{B}(x')$ or $\mathcal{B}(x)$). Since $\mathcal{B}$ is $(\varepsilon_1, \delta_1)$-DP and since post-processing does not affect differential privacy, it follows that for every event $E$

$$p_{x'}(E) \leq e^{\varepsilon_1} p_{\mathrm{mix}}(E) + \delta_1$$

Let $F$ denote the event that $\widetilde{\mathrm{LS}}_x > \mathrm{LS}_f(x)$. By the precondition of the lemma, $\Pr[\mathcal{B}(x) > \mathrm{LS}_f(x)] > 1 - \delta_2$, $\Pr(F) > 1 - \delta_2$. Here, $z \in \mathbb{R}^d$ is an arbitrary point.

We have

$$\frac{p_{\mathrm{mix}}(z|F)}{p_x(z|F)} = \frac{\prod_{i=1}^d e^{-\varepsilon_2|f(x')_i - z_i|/\widetilde{\mathrm{LS}}_x}}{\prod_{i=1}^d e^{-\varepsilon_2|f(x)_i - z_i|/\widetilde{\mathrm{LS}}_x}} = \prod_{i=1}^d e^{\frac{\varepsilon_2(|f(x)_i - z_i| - |f(x')_i - z_i|)}{\widetilde{\mathrm{LS}}_x}}$$

$$\leq \prod_{i=1}^d e^{\frac{\varepsilon_2|f(x) - f(x')|}{\widetilde{\mathrm{LS}}_x}} = e^{\frac{\varepsilon_2\|f(x) - f(x')\|_1}{\widetilde{\mathrm{LS}}_x}} \leq e^{\frac{\varepsilon_2\|f(x) - f(x')\|_1}{\mathrm{LS}_f(x)}} \leq e^{\varepsilon_2}.$$

The first inequality follows from the triangle inequality, the second inequality follows from the definition of event $F$, and the third inequality is due to the definition of local sensitivity, $\mathrm{LS}_f(x) \geq \|f(x) - f(y)\|_1$.

For convenience, we can replace points with events, resulting in $p_{\mathrm{mix}}(E|F) \leq p_x(E|F)$. Since the probability of $F$ is the same under both $p_{\mathrm{mix}}$ and $p_x$, we can strengthen this to $p_{\mathrm{mix}}(E \cap F) \leq e^{\varepsilon_2} p_x(E \cap F)$. Note that $\Pr(\overline{F}) \leq \delta_2$ and thus

$$p_{\mathrm{mix}}(E) \leq p_{\mathrm{mix}}(E \cap F) + p_{\mathrm{mix}}(E \cap \overline{F}) \leq e^{\varepsilon_2} p_x(E \cap F) + p_{\mathrm{mix}}(E \cap \overline{F}) \leq e^{\varepsilon_2} p_x(E) + \delta_2.$$

Because we obtain $p_{x'}(E) \leq e^{\varepsilon_1} p_{\mathrm{mix}}(E) + \delta_1$, we get

$$p_{x'}(E) \leq e^{\varepsilon_1 + \varepsilon_2} p_x(E) + e^{\varepsilon_1}\delta_2 + \delta_1.$$

The inequality is symmetric by the whole proof, as it remains valid when $x'$ is replaced with $x$, ensuring the result holds regardless of the order of $x$ and $x'$. So we prove $\mathcal{A}$ is $(\varepsilon_1 + \varepsilon_2, \delta_1 + e^{\varepsilon_1}\delta_2)$-DP. $\square$

---

**Algorithm 8** APPROXDPRSC $(G, H, Q, \varepsilon, \delta)$     ▷ Approximate DP Range Subgraph Counting Query For Mult-attribute

---

1: **Input:** An $n$-vertex graph $G = (V, E, \mathbf{a})$, a pattern graph $H$, a set of range queries $Q$, and privacy parameter $\varepsilon$, $\delta$.
2: Set $\varepsilon'$ and $\delta'$ such that $\varepsilon = (|E_H| + 1)\varepsilon'$ and $\delta = \delta' + (|E_H| + 1)e^{\varepsilon'}\delta'$.
3: $\widetilde{\mathrm{HS}}_{f_H}(G) =$ Algorithm 7 $(G, H, \varepsilon', \delta')$.
4: $\mathbf{w} = $ PROJMULT $(G, H)$.
5: $\widetilde{T}_1 = $ TREECONSTMULT $(\mathbf{w}, \varepsilon', \widetilde{\mathrm{HS}}_{f_H}(G))$
6: **for** $q \in Q$ **do**
7:     Get $\ell_i$ and $r_i$ according to Definition 3.2 for each dimension of $q$.
8:     **return** Output of Definition B.6 with $\widetilde{T}_1$ and $[\ell_1, r_1] \times \cdots \times [\ell_d, r_d]$.
9: **end for**

---

*Proof of Theorem 2.* We prove the privacy and utility of the algorithm separately.

**Privacy:** We continue to use $\mathbf{w}$ as the vector output of the subgraph projection algorithm (the same as $\mathbf{w}$ in Lemma 3.6 when $d = 1$). We use $\mathbf{w}$ and $\mathbf{w}'$ to denote the different weight vectors formed by graphs $G$ and $G'$, respectively. Recall that $f_H(G)$ is the subgraph counting function for $G$. We have

$$\mathrm{LS}_{\mathbf{w}}(\mathrm{G}) = \max_{\mathbf{w}, \mathbf{w}'} \|\mathbf{w} - \mathbf{w}'\|_1 = \max_{G':G'\sim G} |f_H(G) - f_H(G')| = \mathrm{LS}_{f_H}(\mathrm{G}).$$

Thus, if we get noisy estimate of $\mathrm{LS}_{f_H}(G)$, we get noisy estimate of $\mathrm{LS}_{\mathbf{w}}(G)$. Obviously, we can get $\widetilde{\mathrm{HS}}_{f_H}(G)$ for $(k_H\varepsilon', \delta' + k_H e^{\varepsilon'}\delta')$-DP by Lemma E.1. According to Lemma E.3, if we release $\mathcal{A}(G) = (\widetilde{\mathrm{LS}}_{\mathbf{w}}(G), \mathbf{w} + \mathrm{Lap}(\widetilde{\mathrm{LS}}_{\mathbf{w}}(G)/\varepsilon')) = (\widetilde{\mathrm{HS}}_{f_H}(G), \mathbf{w} + \mathrm{Lap}(\widetilde{\mathrm{HS}}_{f_H}(G)/\varepsilon'))$, we can obtain a $((k_H + 1)\varepsilon', \delta' + (k_H + 1)e^{\varepsilon'}\delta')$ estimate of $\mathbf{w}$.

Note that we are not aiming to obtain a differentially private $\mathbf{w}$; instead, our goal is to ensure that the constructed tree satisfies privacy requirements, as referenced in Lemma 3.6. Let $\mathbf{w}_t$ represent the vector of weights of all nodes in innermost trees (for $d = 1$, this corresponds to all trees $T_y$; for $d \geq 1$, it corresponds to all trees $T_d$). We mention the description of $\mathbf{w}_t$ in Lemma 3.6 when $d = 1$.

The vector $\mathbf{w}_t$ satisfies $\mathrm{LS}_{\mathbf{w}_t} = \mathrm{LS}_{\mathbf{w}} \cdot \log^{2d} n$. The noisy estimate $\widetilde{\mathrm{HS}}_{\mathbf{w}_t}(G)$ is actually $\log^{2d} n$ times the noise estimate $\widetilde{\mathrm{HS}}_{\mathbf{w}}(G)$.

Therefore,

$$(\widetilde{\mathrm{LS}}_{\mathbf{w}_t}, \mathbf{w}_t + \mathrm{Lap}(\widetilde{\mathrm{LS}}_{\mathbf{w}_t}(G)/\varepsilon')) = (\widetilde{\mathrm{HS}}_{\mathbf{w}_t}, \mathbf{w}_t + \mathrm{Lap}(\widetilde{\mathrm{HS}}_{\mathbf{w}_t}(G)/\varepsilon'))$$
$$= (\widetilde{\mathrm{HS}}_{\mathbf{w}} \cdot \log^{2d} n, \mathbf{w}_t + \mathrm{Lap}(\widetilde{\mathrm{HS}}_{\mathbf{w}}(G) \cdot \log^{2d} n/\varepsilon'))$$
$$= (\widetilde{\mathrm{HS}}_{f_H} \cdot \log^{2d} n, \mathbf{w}_t + \mathrm{Lap}(\widetilde{\mathrm{HS}}_{f_H}(G) \cdot \log^{2d} n/\varepsilon'))$$

is $((k_H + 1)\varepsilon', \delta' + (k_H + 1)e^{\varepsilon'}\delta')$ -DP, where $\varepsilon'$ and $\delta'$ is privacy parameter in Algorithm 7. Here, we set

$$\varepsilon = (k_H + 1)\varepsilon', \quad \delta = \delta' + (k_H + 1)e^{\varepsilon'}\delta'.$$

By the post-processing property, Algorithm 8 satisfies $(\varepsilon, \delta)$-DP. Furthermore, if $\varepsilon$ and $\delta$ are specified, $\varepsilon'$ and $\delta'$ can be easily computed.

**Utility:** The overall proof is similar to the utility proof in Theorem 1. Recall that, in Step 5 of Algorithm 8 (which calls Algorithm 5), we add independent Laplace noise with a magnitude of $O(\widetilde{\mathrm{HS}}_{f_H}(G) \cdot \log^{2d} n)$ to the weight of each tree node. For a fixed query $q$, the additive error generated is

$$\left| f_H(G_q) - \widetilde{f}_H(G_q) \right| = \left| \sum_p w(p) - \sum_p \widetilde{w}(p) \right| \leq \left| \sum_p^{\log^{2d} n} Y_p \right| = O\left(\frac{\widetilde{\mathrm{HS}}_{f_H}(G) \cdot d \cdot \log^{3d+0.5} n}{\varepsilon'}\right)$$
$$= O\left(\frac{\widetilde{\mathrm{HS}}_{f_H}(G) \cdot d \cdot \log^{3d+0.5} n}{\varepsilon}\right)$$

With a probability of at least $1 - \frac{1}{n^3}$ (as established in Lemma 2.6), we have $b = \widetilde{HS}_{f_H}(G) \cdot \log^{2d} n$, $m = \log^{2d} n$, and $\beta = n^{2d+1}$, as supported by Lemma D.2, Lemma D.4, and Lemma 2.6. The inequality holds because each query uses at most $\log^{2d} n$ tree node weights for computation, as shown in Lemma D.4. For the final equality, note that we focus on the family of pattern graphs with a constant number of edges, where $k_H$ is a constant. □

However, there is no explicit upper bound on $\widetilde{HS}_{f_H}(G)$ for all $H$, and its value typically varies depending on $H$ and $G$. For some $H$, $\widetilde{HS}_{f_H}$ can be relatively easy to estimate, while for others, it presents more significant challenges. Nevertheless, our results remain practically significant. For common $H$, such as triangles, $\widetilde{HS}_{f_\triangle} \approx d_{\max}(G)$, where $d_{\max}(G)$ represents the maximum degree of the graph $G$. In most sparse graphs in the real world, $d_{\max}(G) = o(n)$.

**Lemma E.4** ((Karwa et al., 2011)). *It holds that*

$$\widetilde{HS}_{f_\triangle} \leq d_{\max}(G) + \frac{2 \ln 1/\delta'}{\varepsilon'}$$

*with probability at least $1 - \delta'$.*

*Proof.* (Karwa et al., 2011) provided a proof for the case of $k$-triangles. For clarity, we have rewritten the proof for triangles.

If $H$ is a triangle, then $f_\triangle^{(1)}(G) \leq d_{\max}(G)$, $f_\triangle^{(2)}(G) = 1$. According to the algorithm Algorithm 7, $\widetilde{HS}_{f_\triangle} = f_\triangle^{(1)}(G) + \frac{\ln 1/\delta'}{\varepsilon'} + \text{Lap}(1/\varepsilon') \leq d_{\max}(G) + \frac{\ln 1/\delta'}{\varepsilon'} + \text{Lap}(1/\varepsilon')$. We have $\widetilde{HS}_{f_\triangle} \leq d_{\max}(G) + \frac{2 \ln 1/\delta'}{\varepsilon'}$ with probability at least $1 - \delta'$ by Fact 2.5. □

## F EDGE-ATTRIBUTED RANGE SUBGRAPH COUNTING PROBLEM

In practical applications, many works require counting subgraphs based on edge attributes. For example, in dynamic graphs, temporal networks or relational event graph with edges that have timestamps, someone want to query the number of subgraphs related to edges generated within a certain time range in order to calculate metrics like clustering coefficients for data mining purposes. Therefore, we have revised our definition and introduced algorithm for range subgraph counting based on edges.

**Definition F.1** (Edge Range Subgraph Counting). *$G = (V, E, \mathbf{a})$ is an undirected graph where each edge $e \in E$ carries a real-valued attribute $\mathbf{a}(e)$. For an interval $q = [\ell_1, r_1] \times \cdots \times [\ell_d, r_d]$, define $E_q = \{e \in E | \ell_i \leq \mathbf{a}_i(e) \leq r_i, i \in [d]\}$ and $G_q$ as the subgraph of $G$ induced by $E_q$.*

*Let $H$ be a connected (pattern) graph with a fixed number of vertices, e.g., triangle, edge, star. Given an interval $q$, a query returns the number of occurrences of $Q$ in $G_q$. The pattern $H$ is fixed for all queries.*

We show that our previous algorithm framework is so powerful that it can be used to solve this problem with a simple adjustment, which also shows the versatility of our algorithm.

To distinguish them from vertices, we use $e^i$ to denote edges. At the beginning, we have the initial labels of the edges. Similarly, we use $s_j(e^i)$ to denote the rank after reordering according to the $j$-th dimension attributes. For edge range subgraph counting, we only need to adjust the projection part, and the rest of the algorithm content will reuse Algorithm 2 and Algorithm 3, just replace vertices with edges.

---

**Algorithm 9** EDGEPROJ($G = (V, E, \mathbf{a}), H$)    ▷ Edge Subgraph Counting Projection

---

1: **Input**: An $n$-vertex graph $G = (V, E, \mathbf{a})$.
2: Reorder all edge labels by attribute value from small to large. If the attribute values are the same, sort according to the initial label. Obtain the new rank $s : E \to [n^2]$.
3: Initialize $w_{(e_1^1, e_1^2, \ldots, e_d^1, e_d^2)} = 0$, for any $e_j^1, e_j^2 \in E$ where $j \in [d]$.
4: **for all** occurrences of subgraph $H$ in $G$ **do**
5:   Compute $w_{(s_1(e_1^1), s_2(e_1^2), \ldots, s_d(e_d^1), s_d(e_d^2))} = w_{(s_1(e_1^1), s_2(e_1^2), \ldots, s_d(e_d^1), s_d(e_d^2))} + 1$, where occurrence registered at $(e_1^1, e_1^2, \ldots, e_d^1, e_d^2)$.
6: **end for**
7: **return** $\mathbf{w} = \{w_{(s_1(e_1^1), s_2(e_1^2), \ldots, s_d(e_d^1), s_d(e_d^2))}\}$

---

Refer to the construction of Algorithm 2 and Algorithm 3, just replace the vertices with edges. The proof follows Section 3. The specific proof process is similar to vertex attribute case, we will not repeat them here for simplicity. The only difference is that here we use edges to determine the range, and there are at most $O(n^2)$ types of edges, so there are at most $O(n^{4d})$ possible queries. For building a DP range tree, $n^{2d}$ tuples are used to build a $d$-dimensional DP range tree, and at most $\log^{2d} n^2$ nodes are used each time. We then have the following theorem.

**Theorem 3** (Pure DP Edge-Attributed Range Subgraph Counting). *For any $\varepsilon > 0$, there exists an $\varepsilon$-DP efficient algorithm that given a graph $G = (V, E, \mathbf{a})$, where the attribute of each **edge** is a $d$-dimensional vector, a pattern graph $H$, and a query set $Q$ outputs all subgraph counting queries which satisfy*

$$\max_{q \in Q} \left| f_H(G_q) - \widetilde{f}_H(G_q) \right| = O\left( \frac{\mathrm{GS}_{f_H} \cdot d \cdot \log^{3d+0.5} n}{\varepsilon} \right)$$

*with probability at least $1 - \frac{1}{n}$.*

**Theorem 4** (Approximate DP Edge-Attributed Range Subgraph Counting). *For any $\varepsilon > 0$ and $0 < \delta < 1$, there exists an $(\varepsilon, \delta)$-DP efficient algorithm that given a graph $G = (V, E, \mathbf{a})$, where the attribute of each **edge** is a $d$-dimensional vector, a pattern graph $H$, and a query set $Q$ outputs all subgraph counting queries which satisfy*

$$\max_{q \in Q} \left| f_H(G_q) - \widetilde{f}_H(G_q) \right| = O\left( \frac{\widetilde{\mathrm{HS}}_{f_H} \cdot d \cdot \log^{3d+0.5} n}{\varepsilon} \right)$$

*with probability at least $1 - \frac{1}{n}$.*

We extend Theorem 2 to the edge case. By performing edge projection using Algorithm 9 and replacing the global sensitivity with $\widetilde{\mathrm{HS}}_{f_H}(G)$ estimated via Algorithm 7 to construct the DP range tree, we achieve an error of $O(\widetilde{\mathrm{HS}}_{f_H}(G))$, ignoring $d$ (since terms involving $d$ remain unchanged). In general, $\widetilde{\mathrm{HS}}_{f_H}(G)$ provides better results than global sensitivity.

