# OpenReview forum: "Differentially Private Range Subgraph Counting"
_ICLR.cc/2025/Conference — Submitted to ICLR 2025_

### Official Review · Reviewer_nCCQ · 2024-11-03

**Soundness:** 3
**Presentation:** 3
**Contribution:** 2
**Rating:** 5
**Confidence:** 4

**Summary:**

The paper is on the problem of private subgraph counting in graph analytics, in particular, the "range subgraph counting" problem.  Here, the set up is that each vertex has numeric attributes (in dimension d) and queries specify intervals that induces appropriate subgraphs.  The goal is to do subgraph counting with DP.  The problem is reasonably motivated and the paper tries to give some examples where it might arise.

Applying DP to this problem naively would hurt utility (due to sensitivity & composition) and not exploiting the query overlap.  The idea in the paper is to leverage the work done in high-dimensional range counting, using trees, which can exploit overlapping queries and hence add lower noise.

The technical idea is to map the number of occurrences of a subgraph to a suitable range in 2d space (here for simplicity, say d=1, ie, each vertex has a scalar numeric attribute).  This way, the problem can be "reduced" to 2d range counting with trees.

The paper provides experimental results in two public datasets.  The experimental results are as expected and do not throw any surprises.

The Appendix contains interesting material about sensitivity and fractional edge cover.  But the main result actually follows in a straightforward way from the result of Atserias et al 2018.

The higher-dimensional extension (d>1) is straightforward.

**Strengths:**

* The paper proposes an interesting set up which is reasonably motivated from both graph analytics and privacy points of view

* The paper has a principled approach to the problem, with theoretical analysis

**Weaknesses:**

* The paper is more of a theory paper with experiments as add-on.  Given this,  it is quite weak from a theoretical point of view.  The mapping (the paper calls is "registered") is itself straightforward and does not offer any new non-trivial insights.

+ The bells and whistles (utility analysis, privacy analysis for the Laplace noise, the tree properties, etc) all follow from standard DP literature.  This makes the novelty aspect of the paper, especially its solution part, low and the work itself unappealing.

**Questions:**

181: should it be a: V -> R?  (Or since you are defining it here, R^d, as you set d=1 only later)

207: X_i ~ Lap(b_i)?

233: what is "rank pair"?

239: what is "bound information"?

243-246: to make exposition clearer, these lines defining being "registered" could be moved earlier, say, before line 232.  Incorporing the definition of s(.) in text (instead of in Alg 1) will also be helpful to avoid going back and forth.
Def 3.1  Is u = u_0, v = u_{|V_H|-1}?  Saying what "registered" at (u, v) means in words might be helpful.  Also giving intuition of why we need this notion (basically some canonical way of mapping an occurrence) will be helpful.

265: This para could be moved much earlier for improved flow.

411: Might be useful to spell out the last step in the equation

Experiments: There are so many publicly available graphs with node attributes.  You didnt ahve to generate vertex attributes synthetically.

Is it possible to open up the subgraph counting algorithms to get a better solution to your problem?

---

> ### Author Response · Authors · 2024-11-21
>
> We sincerely thank the reviewer for their valuable comments and suggestions.
>
> A1(181), A2(207): Thanks. We will fix these typos.
>
> A3(233): A rank pair refers to a pair $(a, b)$, where a and b represent the ranks of a vertex based on its attribute value and index order, respectively. Thanks for pointing out.
>
>  A4(239): Fact 3.4 states that a tree node stores information including the interval bound and weight. The interval bound records the range of attributes (rank) that this tree node controls, which we refer to as the bound information. We will make careful revisions.
>
> A5, A6, A7(243-246,265,411): Thank you very much for your suggestion, we will make careful revisions.
>
> About Experiments: We conducted a thorough investigation on several graph data repositories (e.g., SNAP, Network Repository). Most publicly available graphs with vertex attributes are embedded or anonymized due to privacy concerns, making them unsuitable for range subgraph queries. For example, the Facebook dataset is anonymized and embedded at the same time, node features are one-hot encoded (e.g., $[0, 1, \dots]$) and lack explicit "range". The few graphs that do meet the criteria pose significant challenges for experimentation due to their large scale and equipment limitations, like Pokec Social Network ($n > 10^6$, $m > 10^8$).
>
> About Algorithm: For certain pattern graphs, it may be possible to improve time complexity using more advanced enumeration or counting algorithms. However, one cannot hope to improve too much the utitlity of out algorithm, as we have shown that our algorithm achieves nearly tight additve error, i.e., the global sensitivity of the subgraph counting problem.

---

> > ### Comment · Reviewer_nCCQ · 2024-11-28
> > **Comment**
> >
> > Thank you for your response.  Read & acked.

---

### Official Review · Reviewer_px1m · 2024-11-04

**Soundness:** 4
**Presentation:** 3
**Contribution:** 3
**Rating:** 6
**Confidence:** 4

**Summary:**

Given a graph with each node accompanied by a vector, this paper proposes an algorithm for counting a given graph in the subgraph induced by a rectangular range query. The algorithm has no dependence on the size of the query set; to achieve this, they map each occurrence of the given graph to the rectangle for which the graph exists in space, then use a range tree to perform the suitable aggregation to answer any query rectangle.

**Strengths:**

The algorithm is simple and easy to implement. The paper does a good job explaining it and its intuition. The algorithm is practical for small subgraphs because it adds little runtime overhead on top of counting the subgraphs.

The idea of reducing graph counting in a subset of the original graph to range queries in 2d-dimensional space is elegant.

**Weaknesses:**

Especially in graphs, it is often pessimistic to add noise scaling with the global sensitivity. A simple observation is that the degree of a real-world graph is usually much less than the number of users, and this will make the local sensitivity significantly less. This has been done in prior work [FHO21]; that paper proposes graph algorithms under continual observation, which is very related to this problem setup when d = 1, and it obtains error that scales with the maximum degree of the graph.

The algorithm is less efficient when counting larger graphs (which is to be expected) or when the vector dimension is large.

[FHO21] Fichtenberger, Hendrik, Monika Henzinger, and Lara Ost. "Differentially private algorithms for graphs under continual observation." arXiv preprint arXiv:2106.14756 (2021).

**Questions:**

The paper [FHO21] should be cited. They propose an algorithm for counting triangles; how does it compare to the proposed algorithm (e.g. with d = 1)?

A work-around to the maximum degree issue would be to compute a maximum degree over-estimate privately, and then to run the existing algorithm adding noise tailored to this estimate. How feasible is it to use this or a related idea in order to attain error scaling with the maximum degree instead of with n?

In the experiments, is it realistic to generate the vector for each node from a Gaussian? What is d in the experiments---can it be run with higher d to evaluate how high of a dimension it may be scaled to?

---

> ### Author Response · Authors · 2024-11-21
>
> Thank you for pointing out the reference. After carefully reviewing [FHO21], we acknowledge that there is indeed a connection between their work and our manuscript, as both address differential privacy (DP) algorithms for subgraph pattern counting within certain families of subgraphs. However, the families of subgraphs considered in the two works differ fundamentally.
>
> [FHO21] focuses on the family of subgraphs corresponding to those present at any given timestamp, whereas our work considers the family of subgraphs induced by vertices/edges whose attributes fall within specific ranges.
>
> Comparison of Problems:
>
>    1. Vertex-Attribute Range Subgraph Counting: For this problem, the two families (and hence the two problems) are incomparable.
>
>    2. Edge-Attribute Range Subgraph Counting: Interestingly, we found that our problem serves as a strict generalization of the partially dynamic problem addressed in [FHO21].
>
>    - Take the incremental graph sequence as an example: if we view the timestamp of an edge as its edge attribute, then a query corresponding to the range $[1, T]$ in our problem is equivalent to the graph at timestamp $T$ in the incremental graph problem.
>
>    - In this sense, the partially dynamic problem is a special case of our edge-attribute range subgraph counting problem, with $d=1$ and the edge attribute being its timestamp. Consequently, our Theorem 2 also provides the corresponding DP guarantee for the partially dynamic problem, with an additive error equal to the global sensitivity of subgraph counting.
>
>    3. Pattern Graph $H$:
> Finally, [FHO21] limits its analysis to specific pattern graphs, such as triangles and $k$-stars, while our work considers arbitrary constant-size pattern graphs $H$, making it more general in this regard.
>
> 4. Algorithm:  First, our algorithm uses a tree structure to reduce the amount of noise added, which is similar to [FHO21]. The difference lies in addressing our more general problem (for general pattern graph counting), where we need to perform subgraph projection to  pre-process. Additionally, during the tree construction process, unlike their method that considers only the temporal dimension, we construct a two-dimensional tree when \(d=1\), even for triangle counting, to achieve our goal. This is because, intuitively, our algorithm not only needs to compute results for linear timestamp (e.g., $t=1, t=2, \dots$) but also needs to calculate results over a range (e.g., within attributes $[i, j]$).
>
> For noise scale: We agree with the point that adding a maximum degree for privacy may help reduce the error in triangle counting, and we greatly appreciate this insight being highlighted.
>
> There are two cases here for triangle counting:
>
>     1. The upper bound of the maximum degree of the input graph is known.
>     2. The upper bound of the maximum degree is unknown.
>
> For the first case, if prior knowledge of the upper bound on the maximum degree of the input graph $D$ is available (e.g., as assumed in [FOH21], Lemma 15), the global sensitivity of triangle counting becomes $D$. In such cases, we can achieve an additive error $\widetilde{O}(D)$ for triangle counting and attain pure differential privacy.
>
> For the second case, we have analyzed it and found that it achieves approximate differential privacy (($\varepsilon$,$\delta$)-DP) while guaranteeing an error bound of $\widetilde{O}(d_{max})$ by [KRSY14] lemma 4.4 where $d_{max}$ is the maximum degree of current input graph. Interestingly, the scale of the added noise approaches the local sensitivity and small in most case.  However, it provides strict theoretical guarantees for only a limited number of regular pattern graphs (see [NHSV24], Theorem 3).
>
> For the experiments: In the experiments, we missed providing the setting instructions for $d=1$. We then considered the Gaussian distribution setting to be realistic. First, many real-world vertex attributes, such as height and weight in populations, follow or approximately follow a Gaussian distribution. Secondly, the error is mainly affected by the number of noise terms rather than vertex attributes. The Gaussian distribution of vertex attributes is sufficient from a verification perspective and practical applications. Moreover, the embedding processing of these public graph features, which makes range queries impossible, has also impacted our experiments.
>
> [KRSY14] Karwa, Sofya Raskhodnikova, Adam Smith, Grigory Yaroslavtsev. "Private analysis of graph structure."
> Proceedings of the VLDB Endowment 4.11 (2011): 1146-1157.
>
> [NHSV24] Dung Nguyen, Mahantesh Halappanavar, Venkatesh Srinivasan, Anil Vullikanti. "Faster approximate subgraph counts with privacy." Advances in Neural Information Processing Systems 36 (2024).

---

> ### Comment · Reviewer_px1m · 2024-11-22
> **Thanks for your response**
>
> I agree that the results, with an initial step used to upper bound the local sensitivity, generalize those in the existing paper in [NHSV24].
>
> Assuming a discussion on the steps for estimate the maximum degree is added, I feel more positive about the theoretical contribution of this work, and lean towards acceptance.

---

### Official Review · Reviewer_1Q4n · 2024-11-04

**Soundness:** 3
**Presentation:** 1
**Contribution:** 2
**Rating:** 6
**Confidence:** 3

**Summary:**

This paper studies differentially private subgraph counting, but in a new interactive setting where each time, the algorithm outputs the count of subgraphs in a specific region of the graph induced by vertices. In particular, each vertex is associated with a $d$-dimensional real valued vector, and each query is a $d$-dimensional range query $q = [r_1, l_1],\cdots, [r_d, l_d]$. Then, the goal is to count the occurrences of some pattern $H$ in the subgraph decided by the given range query $q$.

To achieve this goal, they first map the graph to a $2d$-dimension grid based on ranked vertex attributes to translate the range subgraph counting problem into estimating the weighted sum of points in a high-dimensional range. Then, the authors are able to utilize the well-kown constructure of (Bentley & Saxe, 1978) to construct a binary tree for answering range query.

The authors also generalized their results to counting subgraphs based on edge attributes, which similar techniques.

**Strengths:**

1. The subgraph counting problem with differential privacy is interesting and important, and I am glad to see new progress being made in this direction.
2. I believe the problem setting with range queries based on vertex/edge attributes is also kind of practical, although I understand that you may be considering this setting primarily for technical reasons.

**Weaknesses:**

I think there are some places of the construction and the proof of your algorithms that seem somewhat questionable. But this is very likely due to my misunderstanding, so I still would like to give a cautiously positive evaluation. But in any case, I think the way you describe the algorithm makes me very hard to evaluate the real technical contribution of your paper.

In particular, when you highlighting the difference between your work with other DP interval or range query problems, I found the only interesting (and perhaps the only non-trivial) difference to me is that the subgraph counting problem is nonlinear. However, after reading your paper, I do not see how you specifically address the issue of nonlinearity, and thus I think there *may* be some issues about the correctness of the proof (please see "Questions").

**Questions:**

In Fact 3.4 (Properties of Range Tree), you claim that "The **sum** of the values of the tree nodes **equals** the **sum** of the values of the left child **plus** the **sum** of the values of the right child". I believe this is the case in the basic Bentley & Saxe's construction, as well as DP-interval (and rectangle) query. But to me, it is not true when you consider subgraph counting. For example, consider a four-vertex graph where $V = \lbrace 1,2,3,4 \rbrace$ with attributes $s(1)<s(2)<s(3)<s(4)$, and a pattern $H$ where $H$ is a four-length path. Clearly for each range involving less than $4$ vertices (represented by some internal vertex of the tree), the count of $H$ is zero. But the value of the root node can be $1$, right?

In other words, I think the property you described in Fact 3.4 is kind of "linearity", so why you can apply Fact 3.4 to subgraph counting? If my understanding about this issue is correct, then I would instead suggest rejecting this paper. If my understanding about this issue is incorrect, then could you please explain how do you handle the non-linearity?

Actually I think it would be even okay if Fact 3.4 does not hold, as you can always recompute the sum of the values of the internal tree nodes and add Laplace noise, instead of considering it as the sum of its children. I think the real part that I feel there might be issue with the non-linearity is that when the range involving vertices that there is *no* non-leaf node whose value represents the answer of the range query on them. For instance, in the above 4-vertex graph example (which is also your Fig 1), can you describe how you algorithm handle the range query $[s(2), s(3)]$? (I understand how it works when asking $[s(1), s(2)]$, $[s(3), s(4)]$ and $[s(1), s(4)]$ since there are nodes to represent them in your Fig 1.)

---

> ### Author Response · Authors · 2024-11-21
>
> Thank you for your review.
>
> It seems there is some misunderstanding. Specifically, in the range tree we are constructing, the leaf nodes do \textbf{not} correspond to individual graph nodes, nor do the internal nodes represent unions of subgraphs contained in their children.
>
> Let us clarify the algorithm in more detail.
>
> Our range tree (for $d=1$) is constructed over a set $S$ of points in a 2-dimensional Euclidean space, where each point $(a, b) \in S$ has an associated weight. This weight corresponds to the number of subgraphs registered by $a$ and $b$.
>
> In your example, the set $S$ will be $\{(1,1), \dots, (4,4)\}$, with all points initially having a weight of 0, except for $(1,4)$, which has a weight of 1 if the graph $G$ contains a length-four path $H$. Informally, even at the lowest levels of the tree, the information about whether $G$ contains $H$ has already been encoded.
>
> We then construct an x-tree, $T_x$, using the x-coordinates of the points in $S$. For each node in $T_x$, we construct a corresponding y-tree using the y-coordinates of points in the interval associated with that node (see Algorithm 2). The construction of $T_x$ and its y-trees follows the Bentley-Saxe framework, ensuring that the property in Fact 3.4 holds: "The sum of the values of a tree node equals the sum of the values of its left child plus the sum of the values of its right child." Specifically, for any node in the trees, if the point $(1,4)$ is part of the node's range, its weight is 1; otherwise, it is 0.
>
> For a subgraph counting query $[s(2), s(3)]$, we identify the relevant nodes in $T_x$ by querying the range $[2,4] \times [1,3]$. This involves selecting two nodes, $A$ and $B$, in $T_x$, corresponding to $[2,2]$ and $[3,4]$, respectively. For each node $A$ or $B$, we then identify two corresponding nodes in their respective y-trees. Since the point $(1,4)$ does not lie within the queried region, the weights of all involved nodes are 0, and the result of the query is 0.
>
> In other words, the nonlinearity inherent in subgraph counting (i.e., the total number of instances of a pattern graph $H$ in the union of two subgraphs $A$ and $B$ is not necessarily equal to the sum of the instances of $H$ in $A$ and $B$) is effectively addressed during the subgraph counting projection. This is achieved by representing the number of corresponding subgraphs using weighted points by projection. After this projection, the constructed range tree adheres to the linear property described in lemma 3.5. So we can ensure that the sum of the values at any node is equal to the sum of the values at its left and right children described in Fact 3.4.

---

> ### Comment · Reviewer_1Q4n · 2024-11-21
>
> Thanks for your response! I just took a much closer look at your construction of binary trees $T_x$ and $T_y$, and now I agree that you can ensure Fact 3.4 with the projection. But the reason I have such a question is mainly because I lack the intuition that how does the binary tree mechanism work on non-linear problems. Therefore, I am trying to understand your privacy analysis, but I found it hard to follow ---- A quick qustion: in Lemma 3.5, could you please explain why $f_H(G)$ equals to $\sum_{(u,v)} w_{(u,v)}$? According to my understanding of your projection, for example, let $G$ be a $2$-length path with $s(1)<s(2)<s(3)$ and $H$ is an one-length path. Then the projection in my mind would be
> $$
> \left[
> \begin{matrix}
>     0 & 0 & 0 \\\\
>     1 & 0 & 0 \\\\
>     2 & 1 & 0
> \end{matrix}
> \right],
> $$
> that is, the left-bottom position represents the query $[s(1),s(3)]$ of value $2$. In this case, it seems that $f_H(G) = 2$ but $\sum_{(u,v)} w_{(u,v)} = 4$. I guess I must be missing something about how your projection (Alg 1) works.

---

> > ### Author Response · Authors · 2024-11-22
> >
> > Thank you for your comments! In the example you provided, the correct output should be:
> > \\[
> > \\begin{bmatrix}
> > 0 & 0 & 0 \\\\
> > 1 & 0 & 0 \\\\
> > 0 & 1 & 0
> > \\end{bmatrix}
> > \\]
> >
> > We believe the confusion arises from the definition in Definition 3.1 (the definition of "registered at"). We apologize for omitting the condition "$u, v \in V(H)$” in Definition 3.1. This condition ensures that each occurrence of $H$ is registered at a unique pair $u, v$, guaranteeing that each occurrence is counted exactly once. As a result, each length-$1$ path is registered exactly once, which explains why $w(1,2)=w(2,3)=1$, and $w(1,3) = 0$ in your example.
> >
> > In Algorithm 1, we enumerate all occurrences of the pattern graph $H$ and increment $w_{(u,v)}$ by 1 if $H$ is registered at the pair $(u, v)$. Since each occurrence of $H$ is registered at a unique pair $(u, v)$, we can ensure (by Lemma 3.5) that the equality $f_H = \sum_{(u,v)} w_{(u,v)} $ holds.
> >
> > Please let us know if you have further questions.

---

> ### Comment · Reviewer_1Q4n · 2024-11-23
>
> I see, so each occurrence of the pattern will only be counted once in the mapped table, only then you can say that the nonlinearity is addressed during the subgraph counting projection. It makes sense to me. Thank you.
>
> Now I realize that the projection is kind of a crucial step for your algorithm. With this projection, you can essentially reduce the problem to DP region query. I guess the "root" of my concern regarding the correctness of your algorithm lies in the fact that I was misled by your Definition 3.1. If Definition 3.1 is understood **literally**, then the whole algorithm does not seem correct to me, as I thought you must do something magic to solve the non-linearity in the binary tree construction step while I did not see it, and this should be very interesting and might be your real technical contribution because before your paper I do not see how to use the Bentley & Saxe's structure to continuously release values of a non-linear function. But it turns out that I just misunderstood how you do the mapping due to flaws in your description, and the tree construction step then feels somewhat folklore to me.
>
> I said in my initial review text that "the way you describe the algorithm makes me very hard to evaluate the real technical contribution of your paper". And indeed now I feel I overestimated your paper. In particular, to emphasize the contribution of your paper compared to previous similar works in interval or range query problems, you said:
> > we address edge-DP in graphs, whereas (Dwork et al., 2015) focuses on differential privacy in tabular data
>
> I think this can be effectively solved by your projection step, right? I believe it is a sweet trick but not surprising or extremly non-trivial.
>
> >  unlike point counting, our subgraph counting problem is nonlinear
>
> As you said in the rebuttal, this is also "effectively addressed during the subgraph counting projection".
>
> >  a single edge change can affect many mapped points and significantly impact subgraph count
>
> This difference is minor, and it seems you may be overstating its significance, especially since your final error bound includes $GS_{f_H}$ because of this issue.
>
> Overall speaking, I take serious issue with the presentation of this paper, and I am not sure if the technical contribution of this paper merits the acceptance to ICLR. Therefore, I do not feel like to raise my score anymore, but I will stick to the conservative score I initially gave.
>
> Best,
>
>
> Reviewer 1Q4n

---

### Author Response · Authors · 2024-11-28
**To all the reviewers: summary of key changes**

Dear Reviewers,

We appreciate the reviewers for their insightful comments and constructive feedback. To reflect the reviewers' feedback, we have uploaded an updated version of the manuscript, with changes highlighted in blue. The key updates include the following:

1. We corrected several typos.

2. We added additional explanations to enhance the readability of the article and modify some statements to avoid misunderstanding.

3. We introduced a new result on approximate differential privacy, which is more practical for real-world networks, and provided the proof in the Appendix E.

4. We included a discussion in comparison with [FHO21].

---

> ### Comment · Reviewer_1Q4n · 2024-11-28
>
> Dear authors,
>
> I really appreciate the efforts you made in the revision to improve the draft, it helps a lot. While I still do not think your technical contribution is “super” interesting, but it is definitely not an unreasonable fit at ICLR. I would be happy to see this paper accepted.
>
> Reviewer 1Q4n

---

### Meta-Review · Area_Chair_oYi2 · 2024-12-14

**Metareview:**

## Summary of Contributions

This paper studies a setting where there is a graph $G$ and each vertex has a feature vector in $\mathbb{R}^d$; the goal is to count the number of occurrences of a certain subgraph $H$ when restricted to the subgraph of $G$ on the nodes whose vector is in a specific rectangle $R = [\ell_1, r_1] \times \cdots \times [\ell_d, r_d]$. We want to do this for all possible rectangles, and to satisfy edge differential privacy. The main result of this paper is an algorithm with an error $\log^{O(d)} n \cdot GS / \epsilon$ error for the problem where $GS$ denote the global sensitivity of an edge. The rough idea is to observe that, for every occurrence of $H$, we can associate to it a rectangle  $R_H = [\ell^H_1, r^H_1] \times \cdots \times [\ell^H_d, r^H_d]$ such that this occurrence is included in the query $R$ iff $R_H \subseteq R$. This essentially reduces the problem to that of range query in $d$ dimensions, for which an algorithm is well known [Dwork et al., 2014].

## Strengths
- The subgraph counting problem with attributes is interesting and seems to be well motivated.
- The reduction / algorithm is simple and can be implemented in practice.

## Weaknesses
- The reduction itself is relatively straightforward. It is unclear how much novel technical contribution there is here.
- The error depends on the global sensitivity whereas some previous work (in the no-attribute setting) can reduce the error to local sensitivity. (This is addressed somewhat during the rebuttal; see Theorem 2 & Appendix E.)
- The error grows quickly with the number of dimensions $d$, making the algorithm impractical even for moderate $d$. And there is no lower bound given that this is the right dependency.
- The feature vectors are assumed to be public. It is unclear how realistic this is and no discussion is provided with regards to this limitation.

## Recommendation

Given the weaknesses, we believe that the paper's contributions are below the bar for ICLR and we recommend rejection.

**Additional Comments On Reviewer Discussion:**

Reviewer 1Q4n was originally confused about the algorithm and the authors clarified that. Another reviewer (px1m) asked for comparison with previous work [FHO21] which the authors also clarified. Although these answers are completely competent, they do not address the weaknesses listed in the metareview.

---

### Decision · Program_Chairs · 2025-01-22

Reject